# Selective electroreduction of carbon dioxide to methanol on copper selenide nanocatalysts

Dexin Yang[1,2], Qinggong Zhu[1,2], Chunjun Chen[1,2], Huizhen Liu[1,2], Zhimin Liu[1,2], Zhijuan Zhao[1], Xiaoyu Zhang[1], Shoujie Liu[3] & Buxing Han[1,2]

Production of methanol from electrochemical reduction of carbon dioxide is very attractive. However, achieving high Faradaic efficiency with high current density using facile prepared catalysts remains to be a challenge. Herein we report that copper selenide nanocatalysts have outstanding performance for electrochemical reduction of carbon dioxide to methanol, and the current density can be as high as 41.5 mA cm$^{-2}$ with a Faradaic efficiency of 77.6% at a low overpotential of 285 mV. The copper and selenium in the catalysts cooperate very well for the formation of methanol. The current density is higher than those reported up to date with very high Faradaic efficiency for producing methanol. As far as we know, this is the first work for electrochemical reduction of carbon dioxide using copper selenide as the catalyst.

[1] Beijing National Laboratory for Molecular Sciences, Key Laboratory of Colloid and Interface and Thermodynamics, CAS Research/Education Center for Excellence in Molecular Sciences, Institute of Chemistry, Chinese Academy of Sciences, 100190 Beijing, China. [2] School of Chemistry and Chemical Engineering, University of Chinese Academy of Sciences, 100049 Beijing, China. [3] College of Chemistry and Materials Science, Anhui Normal University, 241000 Wuhu, China. Correspondence and requests for materials should be addressed to Q.Z. (email: qgzhu@iccas.ac.cn) or to B.H. (email: hanbx@iccas.ac.cn)

Electroreduction of carbon dioxide ($CO_2$) is a potential strategy to transform the intermittent sources of energy into high-energy chemicals, which can potentially reduce our dependence on fossil fuels and alleviate atmospheric pollution[1–4]. Among the products formed upon electrochemical $CO_2$ reduction, hydrocarbons, and alcohols with high energy density, like methanol, are compatible with existing infrastructures and can substitute for fossil fuels[5–7]. It is known that the electrocatalytic reduction of $CO_2$ to methanol requires intricate six-electron/proton coupling steps and sluggish kinetics[8–11]. Therefore, the reaction usually suffers from low current density, poor selectivity and the large overpotential[9,12–14]. As a result, rational design of highly active and robust electrocatalysts that could generate high current density and high selectivity is critical for large-scale application.

Metal and metal-based catalysts have been used for electroreduction of $CO_2$ to CO, hydrocarbons and alcohols[15,16]. To date, some electrocatalysts, such as precious metal and copper-based catalysts, have shown to be promising for electroreduction of $CO_2$ to methanol[9–12,14,17–22]. Among these materials, Cu has been reported as the promising electrocatalyst that is active and selective for $CO_2$ reduction to hydrocarbons and alcohols. However, the activity and selectivity of bulk Cu for producing methanol are usually low[14,18–20]. Metallic Pd or Pt, Pd-Cu and Mo complexes all have also been employed as catalysts for electrochemical synthesis of methanol[10,12,19,21,22]. In addition, Ru/Ti bimetallic oxide is another promising catalyst for production of methanol[23]. It was reported that Mo-Bi bimetallic chalcogenide ($MoS_2/Bi_2S_3$) could be used as electrocatalyst to promote the reaction[8]. Nevertheless, achieving high current density and Faradaic efficiency (FE) simultaneously for conversion of $CO_2$ to methanol remains to be a challenge, and only a few catalysts reported up to date could reach relatively high current density and selectivity, as shown in Supplementary Table 1. Therefore, designing efficient catalysts to enhance the activity and FE, and reduce the overpotential is very interesting from both scientific and practical viewpoints.

In recent years, nanoscale transition metal oxides (TMOs) and chalcogenides (TMCs) have attracted considerable attention, which have great potential of application in photo-electric devices, lithium-ion batteries, gas sensors, and electrocatalysis[24,25]. For electroreduction of $CO_2$, it is known that metal-based oxides, and sulfides could have good performance[8,26–28]. However, only few studies have reported the use of TMCs, such as metal selenides and tellurides for $CO_2$ electroreduction[29,30]. $WSe_2$ nanoflakes were reported as an efficient catalyst for $CO_2$ electroreduction to CO[29]. Density functional theory (DFT) calculation indicated that molybdenum sulfides and selenides were also possible catalysts for $CO_2$ electroreduction[30], which showed that the intermediates COOH and CHO were more easily adsorbed on the S and Se atoms at the edges than the intermediate CO. Therefore, transition-metal selenides may be a class of promising catalysts for $CO_2$ electroreduction. Copper selenides are interesting materials that form nonstoichiometric (i.e., $Cu_{2−x}Se$) as well as stoichiometric (i.e., CuSe, $Cu_2Se$, $Cu_2Se_3$) phases[31,32]. They also have structure stability and composition-dependent optical/electrical properties[33]. In addition, they are low-cost materials compared with many other materials[25], especially noble metals. The advantage of copper selenide is its multiple oxidation states and high electrical conductivity, which can deliver better electrochemical properties[32]. Moreover, the unsaturated Se atoms along its edges may enhance the number of exposed active sites, electrical conductivity and catalytic activity in $CO_2$ reduction[29,30].

Herein, we report a facile solvothermal synthesis of $Cu_{2−x}Se(y)$ nanocatalysts in diethylenetriamine (DETA, Supplementary Fig. 1 for the structure) and $H_2O$ binary solution, where $y$ represents the volume ratio of DETA and water ($V_{DETA}/V_{H_2O}$), and the value of $x$ is in the range of 0.3 to 0.4, depending on the atom ratio of Cu and Se in the catalysts. The properties of the catalysts, such as size and morphology, are solvent-dependent. The catalysts synthesized in the mixed solvent with $V_{DETA}/V_{H_2O}$ of 1:3 can convert $CO_2$ into methanol with a current density of 41.5 mA cm$^{−2}$ at FE of 77.6%. The current density is higher than those reported up to date with very high methanol selectivity (Supplementary Table 1).

## Results

### Synthesis and characterization of $Cu_{2−x}Se(y)$ nanocatalysts.

The $Cu_{2−x}Se(y)$ nanocatalysts were synthesized by solvent coordination molecular template method[34,35], which is shown schematically in Supplementary Fig. 1. In the synthesis of the catalysts, the positively charged ammonium ions coordinated with Se to incorporate into the neighboring $Cu_{2−x}Se(y)$ nanoparticles. The protonated amine molecules then act as a template, resulting in the new morphology of the $Cu_{2−x}Se(y)$ nanocatalysts[34,35].

It is clearly shown that the $V_{DETA}/V_{H_2O}$ affected the morphology of the $Cu_{2−x}Se(y)$ nanocatalysts considerably (Supplementary Fig. 2). The nanoparticles tended to be granular with increasing content of water in the solvent. In addition, the $Cu_{1.63}Se(1/3)$ nanoparticles synthesized at the $V_{DETA}/V_{H_2O}$ of 1/3 had the smallest size. The detailed characterization results of $Cu_{1.63}Se(1/3)$ nanocatalysts are shown in Fig. 1. The images of scanning electron microscopy (SEM) (Fig. 1a) and transmission electron microscopy (TEM) (Fig. 1b) reveal that the size of the $Cu_{1.63}Se(1/3)$ nanoparticles was ~50 nm, which is consistent with the results obtained from dynamic light scattering (DLS, inset of Fig. 1a). Elemental distribution mappings (inset images of Fig. 1b) analysis further indicated the uniform distribution of Cu (yellow) and Se (blue) atoms in the catalysts. Thermogravimetry (TG) curve is shown in Supplementary Fig. 3. The weight loss of DETA (if incorporated into the nanoparticles) should occur at 280–580°C[36]. It can be seen from the figure that there was no weight loss in the temperature range, indicating that there was no DETA in the catalysts. The high-resolution TEM (HR-TEM) image demonstrated the high crystallinity of nanoparticles (Fig. 1c). Furthermore, X-ray photoelectron spectroscopy (XPS) identified the valence states of Cu and Se in the nanocatalysts. The binding energies of Cu $2p_{3/2}$ and Cu $2p_{1/2}$ (Fig. 1d) were 932.1 and 952.0 eV respectively, and there was a peak at 916.9 eV in the Cu LMM spectrum (Fig. 1e), indicating the existence of Cu(I)[37–39]. In addition, the binding energies of Cu $2p_{3/2}$ and Cu $2p_{1/2}$ at 933.5 and 953.9 eV (Fig. 1d) along with the satellite lines can be assigned to Cu(II)[40]. The double peaks at 55.3 and 54.6 eV (Fig. 1f) in the XPS spectrum were the typical Se 3d binding energy for lattice Se$^{2-}$[37,41]. The atomic ratio of Cu(I) to Cu(II) in the nanocrystal determined by XPS was 3.41. Meanwhile, the atomic ratio of Cu to Se determined by inductively coupled plasma-optical emission spectroscopy (ICP-OES) was 1.64, which is consistent with the value determined by XPS (1.63). In terms of the molecular formula, nonstoichiometric $Cu_{2−x}Se(y)$ could be considered as a mixture of stoichiometric $Cu_2Se$ and CuSe[31] and thus the molecular formula can be estimated as $Cu_{1.63}Se(1/3)$. Similarly, the molecular formulae of the nanocatalysts prepared in the solvents at other $V_{DETA}/V_{H_2O}$ ratios were $Cu_{1.64}Se(0/1)$, $Cu_{1.62}Se(1/5)$, $Cu_{1.63}Se(1/1)$, $Cu_{1.60}Se(3/1)$, and $Cu_{1.61}Se(1/0)$ (Supplementary Table 2), and the HRTEM images and XPS spectra are shown in Supplementary Figs. 4–8. The diffraction peaks of the $Cu_{1.63}Se(1/3)$ nanocatalysts (Fig. 1g) can be assigned to (111), (200), (220), (311), (400), and (331) planes of cubic $Cu_{2−x}Se$ (JCPDS No. 06–0680)[37,42], which is in agreement with the HR-TEM result (Fig. 1c). Meanwhile, the difference of

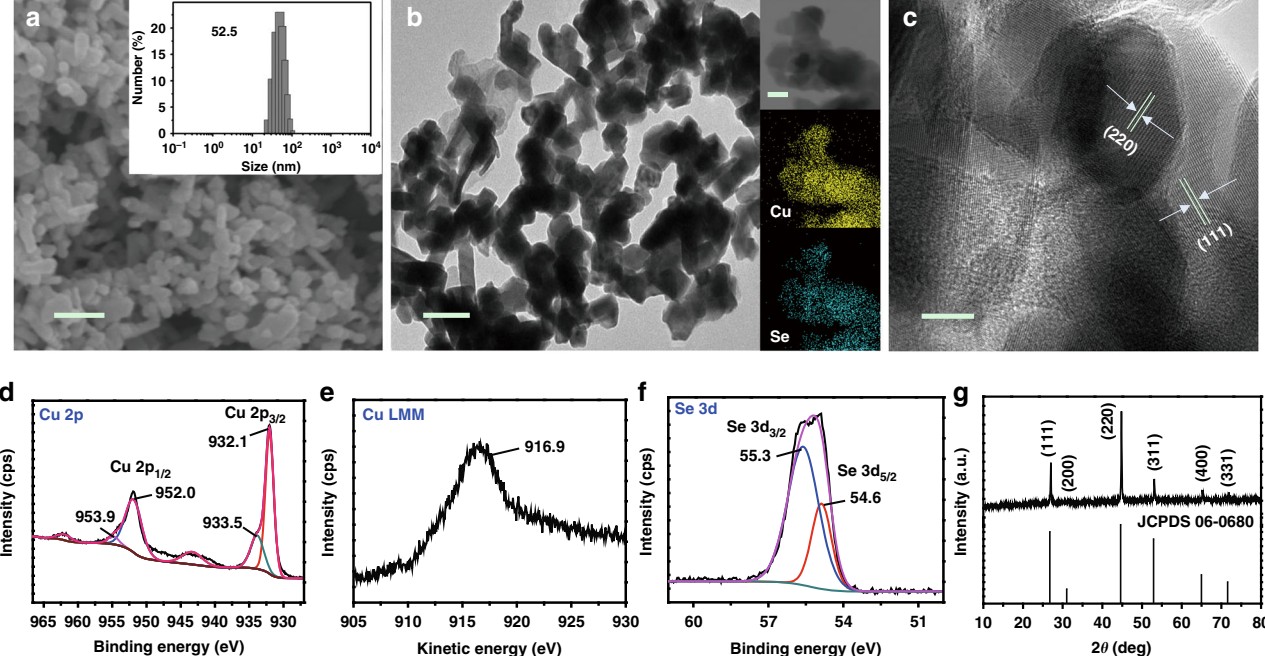

**Fig. 1** Characterization of $Cu_{1.63}Se(1/3)$ nanocatalysts. **a** SEM image of the $Cu_{1.63}Se(1/3)$ nanocatalysts and the inset is size distribution determined by DLS, scale bar = 200 nm; **b** TEM image of the $Cu_{1.63}Se(1/3)$ nanocatalysts and the inset is the corresponding elemental mappings, scale bar = 100 nm; **c** HR-TEM image of the $Cu_{1.63}Se(1/3)$ nanocatalysts, scale bar = 10 nm; XPS spectra of the $Cu_{1.63}Se(1/3)$ nanocatalysts: **d** Cu 2p, **e** Cu LMM, and **f** Se 3d; **g** XRD patterns of the $Cu_{1.63}Se(1/3)$ nanocatalysts

XRD patterns of the catalysts synthesized at various $V_{DETA}/V_{H2O}$ ratios was not noticeable (Supplementary Fig. 9). The results showed that the crystal structures of different samples were not changed with the $V_{DETA}/V_{H2O}$ ratio notably. However, the size and morphology depended strongly on the composition of the solvents, which influenced the performances of $CO_2$ electroreduction.

**Electrocatalytic performance of $CO_2$ reduction over $Cu_{2-x}Se(y)$ nanocatalysts**. The linear sweep voltammetry (LSV) study was conducted to investigate the performances of the $Cu_{2-x}Se(y)$ nanocatalysts under the same conditions in $[Bmim]PF_6$ (30 wt %)/$CH_3CN/H_2O$ (5 wt%) ternary electrolyte, and the results are shown in Fig. 2a. The results indicate that $Cu_{1.63}Se(1/3)$ exhibited a more positive onset potential of −1.815 V vs. $Ag/Ag^+$ than other $Cu_{2-x}Se(y)$ nanocatalysts, suggesting that $Cu_{1.63}Se(1/3)$ was favorable to the binding of $CO_2$. In addition, the current density over $Cu_{1.63}Se(1/3)$ reached a high value of about 40 mA $cm^{-2}$ at −2.1 V vs. $Ag/Ag^+$, which exhibits higher activity than other $Cu_{2-x}Se(y)$ nanocatalysts. The much higher current density of the $CO_2$-saturated than the $N_2$-saturated on the $Cu_{1.63}Se(1/3)$ (around −2.1 V vs. $Ag/Ag^+$) indicates the reduction of $CO_2$.

Constant-potential electrolysis of $CO_2$ over different catalysts was performed in a typical H-type cell[8]. Liquid-phase and gas-phase products were quantified by nuclear magnetic resonance spectroscopy ($^1$H NMR) and gas chromatography (GC), respectively. The current density and FE are displayed in Fig. 2b, c. It can be found that all $Cu_{2-x}Se(y)$ nanocatalysts basically yielded a certain amount of methanol, HCOOH, CO, and $H_2$ with a combined FE of around 100%, and no other product was detected (Supplementary Fig. 10). The $Cu_{1.63}Se(1/3)$ electrode had better performance than other $Cu_{2-x}Se(y)$ nanocatalysts. The maximum FE occurred at −2.1 V vs. $Ag/Ag^+$, and it could reach 77.6% with a current density of 41.5 mA $cm^{-2}$ (Fig. 2b, c). The equilibrium (thermodynamic) potential for $CH_3OH$ was −1.815 V vs. $Ag/Ag^+$, which was obtained by extrapolation of partial current density vs. potential curve to zero partial current density (Supplementary

Fig. 11)[43–45]. Therefore, the overpotential for $CO_2$ electroreduction to methanol was 285 mV at −2.1 V vs. $Ag/Ag^+$. This catalyst exhibits the highest current density with very high FE for producing methanol, as can be known in Supplementary Table 1. The cell voltage is an important factor for practical application, which depends mainly on the performances of the electrocatalysts. In this study, we calculated the cell voltage using the reported method[46,47], and the cell voltage of our system was 2.67 V, which is in the range of reported values (2.2–3.7 V, Supplementary Table 3). In Supplementary Fig. 12, the FE for methanol production increased with the cell voltage to reach the maximum value of 77.6% at 2.67 V. We also calculated the energy efficiency (EE) for methanol production at different cell voltages using the reported method[48], and the results are given in Supplementary Fig. 12. The EE exhibited a similar tendency to the FE of methanol with variation of the cell voltage. Furthermore, the highest EE was 61.7% at the optimized cell voltage of 2.67 V.

We also carried out the experiment using $^{13}CO_2$ to replace $CO_2$ in the electrolysis over $Cu_{1.63}Se(1/3)$. The $^1$H NMR spectra of the product indicated that only $^{13}CH_3OH$ was produced (Supplementary Fig. 13), confirming that the product methanol originated from $CO_2$. Meanwhile, the electrolyte after different electrolysis times was also tested by IR[8,49] in Supplementary Fig. 14, which can further confirm formation of methanol. In addition, FE and current density did not vary during 25 h of operation (Supplementary Fig. 15), which indicated exhibit long-term stability in the electrolysis. The composition and structure of the catalyst did not change after the reaction as characterized by XPS and XRD analysis (Supplementary Figs. 16 and 17), indicating the excellent stability of the $Cu_{1.63}Se(1/3)$.

The particle size and morphology of the materials can affect the number of exposed active sites. From Fig. 2d and Supplementary Fig. 1, we can see that the size of the catalysts synthesized at $V_{DETA}/V_{H2O} = 1/3$ was smallest and the current density and FE were largest. More active sites were exposed with more unsaturated Se atoms on the smaller particles may be one of the main reasons for high current density and FE[29,30].

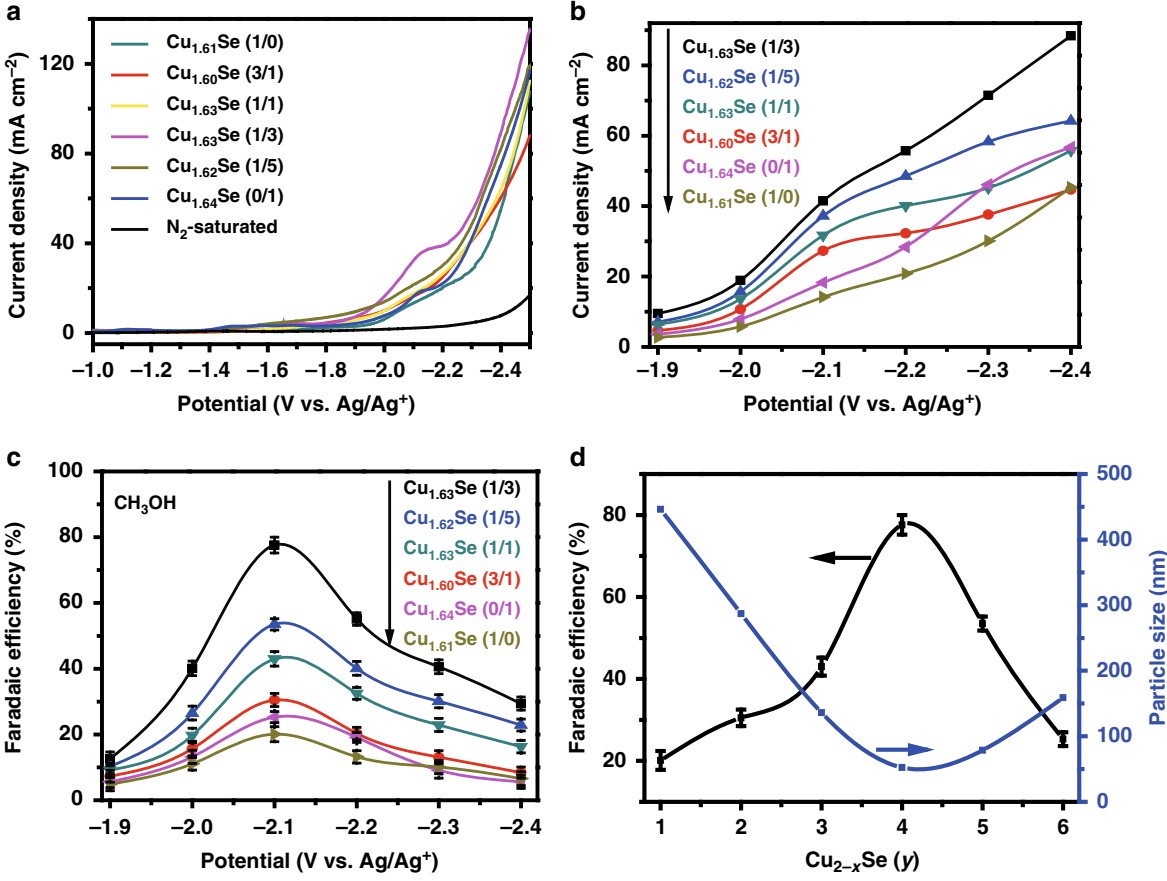

**Fig. 2** $CO_2$ reduction performance on $Cu_{2-x}Se(y)$ nanocatalysts. **a** LSV traces on different electrodes in $CO_2$-saturated or $N_2$-saturated electrolyte at scan rate of 20 mV s$^{-1}$; **b** Total current density and **c** FE over $Cu_{1.63}Se(1/3)$ catalyst at different applied potentials; **d** Plots of particle size vs. FE of methanol on different catalysts (1) $Cu_{1.61}Se(1/0)$; (2) $Cu_{1.60}Se(3/1)$; (3) $Cu_{1.63}Se(1/1)$; (4) $Cu_{1.63}Se(1/3)$; (5) $Cu_{1.62}Se(1/5)$; (6) $Cu_{1.64}Se(0/1)$. Data were obtained at ambient temperature and pressure with $CO_2$ stream of 10 sccm with 5 h electrolysis. All data in **c** and **d** are presented as mean ± s.d

## Discussion

The Tafel plots in Supplementary Fig. 18 and Supplementary Table 2 show the variation of overpotential with partial current density for methanol production over these catalysts. The resulting Tafel plots of various $Cu_{2-x}Se(y)$ electrodes are linear in the overpotential ($\eta$) range from 0.035 to 0.285 V. The Tafel slope of $Cu_{1.63}Se(1/3)$ was smaller than other $Cu_{2-x}Se(y)$ nanocatalysts, which leads to faster increment of $CO_2$ reduction rate with increasing overpotential[26,50]. Combining with efficient and stable electrocatalytic $CO_2$ conversion to methanol, the $Cu_{1.63}Se(1/3)$ nanocatalysts can be a promising catalyst in practical applications.

The excellent activity of $Cu_{2-x}Se(y)$ electrodes may also result partially from large electrochemical active surface area. According to the Randles–Sevcik equation, the current density at $-1.85$ V (vs. Ag/Ag$^+$) plotted against the square root slope of scan rate is shown in Supplementary Fig. 19. The double-layer capacitances ($C_{dl}$) of various catalysts were then calculated and illustrated in Supplementary Table 2. The obvious difference in $C_{dl}$ values (ranging from 0.00906 to 0.0183 Fcm$^{-2}$) suggests that the $Cu_{1.63}Se(1/3)$ had the largest electrochemistry surface area, which is beneficial to the reaction.

Electrochemical impedance spectroscopy (EIS) also revealed an easier electron transfer process on the electrode interface. The Nyquist plot was conducted by running the experiment at an open circuit potential (Supplementary Fig. 20), and the equivalent circuit R(C(R(Q(RW)))) (Supplementary Fig. 21) is then used to

fit the impedance data. The simulated results of charge transfer resistance ($R_{ct}$) are given in Supplementary Table 2. The results confirm that the charge transfer could easily occur on the $Cu_{1.63}Se(1/3)$ electrode. It ensures a faster electron transfer to $CO_2$ for stabilizing reduced $CO_2^{\bullet-}$ intermediate that is vital for electroreduction $CO_2$. All the above results indicate that the particle sizes and morphologies of the catalysts played an important role for electroreduction $CO_2$.

We also used [Bmim]PF$_6$/CH$_3$CN/H$_2$O with different compositions as supporting electrolytes to perform the reaction over $Cu_{1.63}Se(1/3)$. Compared with aqueous electrolyte, combination of organic solvents and ionic liquids as electrolytes has obvious advantages. For example, they can accelerate catalytic performance of $CO_2$ reduction by increasing the adsorption rate of $CO_2$[51,52], and they provide more opportunity to produce various valuable products[1,2,8,53,54]. Meanwhile, the ionic liquids (ILs) with imidazolium cation and fluorine-containing anions can be the efficient supporting electrolytes for $CO_2$ reduction[53,54]. From Supplementary Tables 4 and 5, we can find that the [Bmim]PF$_6$ (30 wt%)/CH$_3$CN/H$_2$O (5 wt%) ternary electrolyte was most efficient for $CO_2$ for the reaction, and the current density could reach 41.5 mA cm$^{-2}$ with a Faradaic efficiency of 77.6%. To further understand the role of anions in the ILs, other ILs were also used, including [Bmim]PF$_6$, [Bmim]BF$_4$, [Bmim]TF$_2$N, [Bmim]OAc, [Bmim]NO$_3$, and [Bmim]ClO$_4$ and acetonitrile systems for comparison (Supplementary Figs. 22 and 23). It can be observed that the anions of the ILs also influenced the

electrochemical reaction significantly (Supplementary Fig. 23), which resulted partially from the difference of the interaction between $CO_2$ and the anions of the ILs[55]. [Bmim]$PF_6$ exhibited higher current density and Faradaic efficiency for methanol among all the ILs used. The separation of the reaction mixture is crucial for practical application. Although this is out of the scope of this work, we would like to discuss this very briefly. For this system, the boiling point of [Bmim]$PF_6$ is much higher than that of $CH_3OH$, $CH_3CN$ and $H_2O$, and the IL in the system can be separated via distillation. Meanwhile, the method to separate ternary mixture consisting of $CH_3OH$, $CH_3CN$ and $H_2O$ has been reported[56].

It is very interesting to investigate the reasons for the outstanding performance of the $Cu_{1.63}Se(1/3)$ electrocatalyst in the electrocatalytic reduction of $CO_2$ to methanol. Therefore, we carried out a series of control experiments to investigate the crucial role of Se in the catalysts. $CO_2$ electroreduction with Cu, CuO, $Cu_2O$, CuS, $Cu_2S$, CuSe, and $Cu_2Se$ as catalysts were studied. Both current density and FE for methanol over Se-free catalysts were obvious lower (Fig. 3 and Supplementary Fig. 24). The results suggest that the Cu and Se in the catalysts cooperated very well for the formation of methanol. In other words, the capacity of electroreduction of $CO_2$ to methanol was enhanced when O or S atom was replaced by Se atom in the catalysts. Moreover, when commercial CuSe or $Cu_2Se$ was utilized as the catalysts, both current density and FE were much lower than that over $Cu_{1.63}Se(1/3)$. The catalytic performance was also evaluated using the electrochemical active surface areas (ECSA) determined by reported method[57]. Results in Supplementary Fig. 25 and Supplementary Table 6 show that the formation rate of methanol over $Cu_{1.63}Se(1/3)$ was intrinsically higher than that on the other catalysts. Thus, on the basis of above results, we can deduce that Se in the catalysts is crucial for efficient $CO_2$ reduction to methanol.

We also carried out extended X-ray absorption fine structure spectroscopy (EXAFS) experiments to study Cu K-edge, which can disclose the local atomic arrangements of the catalysts. The Cu K-edge $k^2\chi(k)$ oscillation curve for $Cu_{1.63}Se(1/3)$ was obviously different from that for CuSe and $Cu_2Se$ (Supplementary Fig. 26). It can be seen that the coordination number in $Cu_{1.63}Se$ (1/3) was smaller than that in CuSe and $Cu_2Se$ (Supplementary Figs. 27–30 and Supplementary Table 7). Thus, there existed unsaturated Se atom in the $Cu_{1.63}Se(1/3)$, which may enhance the performance for $CO_2$ electroreduction.

To understand the reaction pathway for the formation of methanol, some control experiments were conducted in the presence of the possible reaction intermediates, such as formic acid, CO and formaldehyde (Supplementary Table 8). From the production rates of methanol, it can be seen that CO and formaldehyde clearly promoted the formation of methanol and thus they are possible intermediates in the formation of methanol. On the basis of the results above, we propose a possible reaction pathway over $Cu_{2-x}Se(y)$ nanocatalysts (Fig. 4a). In the initial stage of the reduction, the electrolyte containing ionic liquids can enhance the concentration of $CO_2$ in electrolyte and transport of $CO_2$ to the catalyst surface to improve further transform of $CO_2$ into adsorbed $CO_2^{\bullet-}$[2,55]. The adsorbed $CO_2^{\bullet-}$ could bind with the active sites on the surface of catalysts and accelerate the formation of adsorbed-CO species, which was a crucial intermediate for facilitating methanol production as reported[20,58]. Furthermore, the appropriate Cu active sites over the catalysts can enhance the initial intermediate of absorbed-CO to accept electrons and protons to form absorbed-CHO and then reduced to methanol.

The density functional theory (DFT) calculations were also conducted on the multiple elementary reaction steps, and the results are shown in Fig. 4b and Supplementary Figs. 31–33. Comparing with other two catalysts ($Cu_2Se$ and CuSe), the formation of intermediate (*COOH) on the $Cu_{1.63}Se(1/3)$ surfaces can reach a stable configuration with lower free energy via two neighboring Cu atoms through Cu–C and Cu–O bonds. The *COOH intermediate binds with the active sites on the surface of catalysts and accelerate the formation of adsorbed *CO species[59]. The $Cu_{1.63}Se(1/3)$ catalyst also has a moderate binding energy for *CO among the three catalysts, which is beneficial for $CO_2$ transformation to more reduced products that require more than a two-electron reduction[20,58]. Based on the Brønsted–Evans–Polanyi (BEP) relationship[60,61], the reaction barrier has a linear relationship to the reaction energy, and it can also be seen that the step of *CO reduction to *CHO was an endothermic and likely rate-limiting step since the highest energy potential (0.56 eV) is needed in this step. Compared with $Cu_2Se$ and CuSe, the free energy of *CHO over $Cu_{1.63}Se(1/3)$ catalyst is more negative, which may be mainly originated from the moderately strong binding energy for *CO intermediate. In addition, the C–Cu bond (Supplementary Fig. 34) between $Cu_{1.63}Se(1/3)$-CHO is 1.926 Å, which is shorter than those of $Cu_2Se$-CHO (2.188 Å) and CuSe-CHO (2.002 Å), indicating that *CHO is easier to adsorb on the surface of the catalyst to accept electrons and protons to form *$OCH_2$ and *$OCH_3$, and then is reduced to methanol. These results illustrate that the structure distortion of $Cu_{1.63}Se(1/3)$ was beneficial for $CO_2$ electroreduction to methanol.

In summary, a series of $Cu_{2-x}Se(y)$ nanocatalysts were synthesized for selective electroreduction of $CO_2$ to methanol. The Cu and Se in the catalysts had excellent cooperative effect for catalyzing the reaction. The size and morphology were crucial for the performance of the catalysts, which could be controlled by the $V_{DETA}/V_{H2O}$ ratios. The $Cu_{1.63}Se(1/3)$ nanocatalysts yielded outstanding current density of 41.5 mA cm$^{-2}$ with FE of 77.6% at −2.1 V vs. Ag/Ag$^+$. Moreover, the catalyst was also very stable in the reaction. Despite the catalytic system is far from industrial production, it is still very interesting that $Cu_{1.63}Se(1/3)$ nanocatalysts can yield highest current density up to date at very high Faradaic efficiency. We believe that some other transition metal selenides can be designed as efficient electrocatalysts for $CO_2$ reduction.

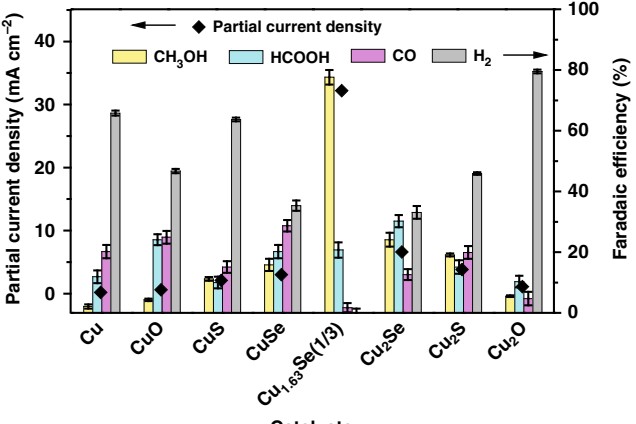

**Fig. 3** Catalytic activity of various Cu-based catalysts. Partial current density and FE of methanol over various catalysts at −2.1 V vs. Ag/Ag$^+$. All data are presented as mean ± s. d

## Methods
**Materials.** CuCl$_2$·2H$_2$O, sulfuric acid (95–98%), ethanol, acetone and acetonitrile ($CH_3CN$) were obtained from Sinopharm Chem. Reagent Co. Ltd. Na$_2$SeO$_3$, diethylenetriamine (DETA), Cu, CuO, Cu$_2$O, CuS, Cu$_2$S, CuSe, Cu$_2$Se,

**Fig. 4** Mechanism study of $CO_2$ reduction to methanol. **a** Proposed mechanism on $Cu_{2-x}Se(y)$ electrode; **b** free energy diagrams on $Cu_{1.63}Se(1/3)$ electrode

hydroxylamine, tetraethylammonium hexafluorophosphate (TEAPF$_6$, purity > 98%), Toray carbon paper (CP, TGP-H-60, 19 × 19 cm), Nafion N-117 membrane (0.180 mm thick, ≥0.90 meg/g exchange capacity) and Nafion D-521 dispersion (5% w/w in water and 1-propanol, ≥0.92 meg/g exchange capacity) were purchased from Alfa Aesar China Co., Ltd. 1-butyl-3-methylimidazolium hexafluorophosphate ([Bmim]PF$_6$, purity > 99%), 1-butyl-3-methylimidazolium tetrafluoroborate ([Bmim]BF$_4$, purity > 99%), 1-butyl-3-methylimidazolium bis(trifluoromethyl sulfonyl)imide ([Bmim]TF$_2$N, purity > 99%), 1-butyl-3-methylimidazolium acetate ([Bmim]OAc, purity > 99%), 1-butyl-3-methylimidazolium nitrate ([Bmim]NO$_3$, purity > 99%) and 1-butyl-3-methylimidazolium perchlorate ([Bmim]ClO$_4$, purity > 99%) were obtained from the Centre of Green Chemistry and Catalysis, Lanzhou Institute of Chemical Physics, Chinese Academy of Sciences. N$_2$, CO$_2$ (99.999%) and $^{13}CO_2$ (99.99%) were provided by Beijing Analytical Instrument Company.

**Synthetic procedures for $Cu_{2-x}Se(y)$ nanocatalysts**. The $Cu_{2-x}Se(y)$ nanocatalysts were prepared according to the procedures reported by other researcher[35]. In a typical procedure, 1 mmol CuCl$_2$·2H$_2$O, 1 mmol Na$_2$SeO$_3$, and 2 mL hydroxylamine were added into a mixed solvent (80 mL) including DETA and deionized water with a volume ratio of $V_{DETA}/V_{H_2O}$ = 1/3 (or 1/0, 3/1, 1/1, 1/5 and 0/1). After stirring for half an hour at room temperature, the blue solution was then transferred into a Teflon-lined autoclave. Subsequently, the sealed Teflon-lined autoclave was maintained at 180 °C for 15 h and then naturally cooled to room temperature. The resulting precipitates were obtained via the method of centrifugation and washed with distilled water and absolute ethanol five times, followed by drying at 80 °C overnight in a vacuum oven.

**Physicochemical characterization**. The microstructures of the catalysts were characterized by scanning electron microscope (SEM, HITACHI S-4800) and transmission electron microscopy (TEM, JEOL JEM-2100F) equipped with energy dispersive spectrometer (EDS). Dynamic light scattering (DLS) measurements in ethanol were performed on a Zetasizer Nano instrument (Malvern Instruments, Worcestershire, UK). X-ray photoelectron spectroscopy (XPS) study was carried out on the Thermo Scientific ESCALab 250Xi using a 200 W Al-Kα radiation. In the analysis chamber, the base pressure was about $3 \times 10^{-10}$ mbar. Typically, the hydrocarbon C1s line at 284.8 eV from adventitious carbon was used for energy referencing. X-Ray diffraction (XRD) analysis of the samples were performed on a Rigaku D/max-2500 X-ray diffractometer with Cu-Kα radiation ($y = 0.15406$ nm) and the scan speed was 5°/min. The elemental contents of the catalysts were detected using inductively coupled plasma optical emission spectroscopy (ICP-AES, Vista-MPX). The thermogravimetric (TG) curve was obtained using Pyris1 TGA under N$_2$ atmosphere.

**Electrode preparation**. The electrode of $Cu_{2-x}Se(y)$/CP was prepared as follow[19]. 10 mg $Cu_{2-x}Se(y)$ prepared above and 1 mg carbon black (Vulcan XC 72) were suspended in the solution with 3 mL acetone and 20 μL Nafion D-521 dispersion (5 wt%) via ultrasound. Then, 302 μL of the above solution was uniformly spread onto the CP ($1 \times 1$ cm$^{-2}$) surface assisted by a micropipette and then dried under room temperature. On each carbon paper, the loading of $Cu_{2-x}Se(y)$ catalyst was 1.0 mg cm$^{-2}$. Before experiment, all the auxiliary electrodes were sonicated in acetone for 10 min and then washed with H$_2$O and absolute ethanol, and then dried in N$_2$ atmosphere.

**Electrochemical study**. All the electrochemical experiments were conducted on the electrochemical workstation (CHI 6081E, Shanghai CH Instruments Co., China). Linear sweep voltammetric (LSV) scans were conducted in a single compartment cell with a three electrodes configuration, including a working electrode, a counter electrode (Pt gauzes), and a reference electrode (Ag/Ag$^+$ with 0.01 M AgNO$_3$ in 0.1 M TBAP-CH$_3$CN). The electrolytes were bubbled with CO$_2$ or N$_2$ at least 30 min to ensure formation of N$_2$-saturated or CO$_2$-saturated solution before experiments. LSV measurements in gas-saturated electrolytes were carried out in

the potential range of −1.0 to −2.5 V versus Ag/Ag$^+$ at a sweep rate of 20 mVs$^{-1}$. Slight magnetic stirring was employed to acquire uniform electrolytes.

**Electrochemical impedance spectroscopy (EIS) study**. The EIS measurement was carried out in [Bmim]PF$_6$-CH$_3$CN-H$_2$O that contents were 30 wt%, 65 wt%, 5 wt% at an open circuit potential (OCP) with an amplitude of 5 mV of $10^{-2}$ to $10^5$ Hz. The data obtained from the EIS measurements were fitted using the software of Zview (Version 3.1, Scribner Associates, USA).

**$CO_2$ reduction electrolysis**. The electrolysis experiments were measured at 25 °C in a commonly used H-type cell, including a working cathode ($Cu_{2-x}Se(y)$/CP), a counter anode (platinum gauzes), and a reference electrode (Ag/Ag$^+$ with 0.01 M AgNO$_3$ in 0.1 M TBAP-CH$_3$CN)[8]. In the experiments, Nafion-117 membrane was used as proton exchange membrane to separate the cathode and anode compartments. 0.5 M H$_2$SO$_4$ aqueous solution and [Bmim]PF$_6$-CH$_3$CN-H$_2$O served as anodic and cathodic electrolytes, respectively. In each experiment, the amount of anodic and cathodic electrolytes was 30 mL. Before starting the electrolysis experiment, the electrolytes were bubbled with CO$_2$ for 30 min under stirring and the electrolysis was carried out under a steady stream of CO$_2$ (10 sccm).

**Product analysis**. After electrolysis reaction, the gaseous products were collected using a gas bag and then analyzed by an Agilent 4890 gas chromatograph equipped with a TCD detector with helium as internal standard. The liquid products were analyzed by $^1H$ NMR measured on a Bruker Avance III 400 HD spectrometer in CD$_3$CN with TMS as internal standard. The Faradaic efficiency of the products was calculated through GC and NMR analysis[8].

**Tafel analysis**. The partial current densities for products under different potentials were measured and the equilibrium potential was obtained by extrapolation method. The overpotential was obtained from the difference between the equilibrium potential and the catalytic potential. Multiple electrolysis experiments were performed at each potential to obtain the current density versus overpotential data in the H-type electrolysis cell as described above. Tafel plots were constructed from these data.

**Double-layer capacitance ($C_{dl}$) measurements**. The electrochemical active surface area is proportional to $C_{dl}$ value. $C_{dl}$ was determined in H-type electrolysis cell by measuring the capacitive current associated with double-layer charging from the scan-rate dependence of cyclic voltammogram (CV). The CV ranged from −1.8 to −1.9 V vs. Ag/Ag$^+$. The $C_{dl}$ was estimated by plotting the $\Delta j$ ($j_a - j_c$) at −1.85 V vs. Ag/Ag$^+$ against the scan rates, in which the $j_a$ and $j_c$ were the anodic and cathodic current density, respectively.

**IR spectroscopy study**. To further understand the process of CO$_2$ electroreduction, a Bruker Tensor 27 IR spectrometer was used to analyze the species produced in the electrolyte. In the experiment, 100 μL electrolyte after desired electrolysis time was dropped on CaF$_2$ disc window and then IR spectrum was obtained.

**Electrochemical active surface areas (ECSA) measurement**. The ECSA values of all electrodes were evaluated by cyclic voltammetry (CV) using the ferri-/ferrocyanide redox couple ([Fe(CN)$_6$]$^{3-/4-}$) as a probe[57]. The CV curves were obtained in a N$_2$-saturated 5 mM K$_4$Fe(CN)$_6$/0.1 M KCl solution including a counter anode (platinum gauze), and a reference electrode (Ag/AgCl with saturated KCl). According to the Randles–Sevcik equation[57], the values of ECSA were obtained.

**The calculation of overpotential ($\eta$) and equilibrium potential ($E^0$)**. Overpotential ($\eta$) is the difference between the equilibrium potential and the actual potential for the transformation of the substrate CO$_2$ into the product methanol via

Eq. (1):

$$\eta = E - E^0_{CO2 \rightarrow methanol} \quad (1)$$

Here, the $E^0_{CO2 \rightarrow methanol}$ referred to the equilibrium potential for $CO_2$ transformation to $CH_3OH$, which can be obtained by extrapolation method[43–45]. Taking the $Cu_{1.63}Se(1/3)$ electrode as example, stepped potential electrolysis experiments between −1.8 and −2.0 V were carried out and the electrolysis products were collected and characterized. The current densities for $CH_3OH$ at each potential are shown in Supplementary Fig. 11, and the potential at $j_{CH3OH} = 0$ by extrapolation method is the equilibrium potential. Therefore, the overpotential can be obtained. The method to calculate the overpotential over other electrodes was similar.

**The calculation of cell voltage**. In this study, we calculated the cell voltage using the method reported[46,47]. It is mainly from the half reaction potentials for water oxidation, $CO_2$ reduction and the ohmic drop ($E_{iR}$) from electrolyte resistance ($R_s$). The $R_s$ stands for solution resistance which was determined by electrochemical impedance spectroscopy (EIS) at frequencies ranging from $10^{-2}$ to $10^5$ Hz and the $I$ represents for amps of average current. Therefore, we calculated $E_{iR}$ by Eq. (2). The applied potentials measured against $Ag/Ag^+$ can be transformed to the reversible hydrogen electrode (RHE) scale by Eq. (3).

$$E_{iR}(vs. RHE) = R_s \times I \text{ (amps of average current)} \quad (2)$$

$$E(vs.RHE) = E(vs. Ag/Ag^+) + 0.54\,V + 0.0591 \times pH. \quad (3)$$

**Cell efficiency**. A characteristic cell energy efficiency (EE) of reduction $CO_2$ to methanol was obtained by Eq. (4)[48]:

$$EE = \frac{FE(\%) \times \Delta E^0}{\text{applied cell voltage}} \quad (4)$$

In the equation, $\Delta E°$ represents the difference between the standard half reaction potentials for water oxidation (1.23 V vs. RHE) and reduction $CO_2$ to methanol (−0.89 V vs. RHE).

**Extended X-ray absorption fine structure (EXAFS) experimental details**. The homogeneously mixed samples (20 mg) and graphite (100 mg) sample were pressed into circular slices with a diameter of 10 mm which was used for further EXAFS measurement under ambient condition. The EXAFS measurements were carried out on the beamline 1W1B station of Beijing Synchrotron Radiation Facility, P.R. China (BSRF). A water-cooled Si (111) double-crystal monochromator (DCM) was utilized to monochromatize the X-ray beam and the detuning was done by 10% to remove harmonics. The electron storage ring of BSRF was operated at 2.5 GeV with a maximum current of 250 mA. The EXAFS data of Cu K-edge was obtained in the energy range from −200 to 1000 eV in transmission mode. The EXAFS oscillations were then extracted according to standard procedures via the ATHENA module implemented in the IFEFFIT software packages. With a Fourier transform k-space range of 2.2–12.8 Å$^{-1}$, the quantitative curve-fittings were conducted in the R-space according to the module ARTEMIS of IFEFFIT software packages. The phase shift $\Phi(k)$ and backscattering amplitude $F(k)$ were calculated by FEFF 8.0 code.

**Computational method**. The free energies of $CO_2$ reduction states were performed using Vienna Ab-initio Simulation Package (VASP)[62], taking advantage of the density functional theory (DFT) and the Projected Augmented Wave (PAW) method[63]. The revised Perdew–Burke–Ernzerhof (RPBE) functional was used to describe the exchange and correlation effects[64,65]. For all the geometry optimizations, the cutoff energy was set to be 450 eV. The (110) surface was chosen to represent the catalysis surface of CuSe; while the (220) surface was modeled to simulate the sites on $Cu_{1.63}Se(1/3)$ and $Cu_2Se$. The Monkhorst-Pack grid[66] of 2 × 2 × 1, 3 × 4 × 1, and 3 × 4 × 1 were used to carry out the surface calculations on CuSe, $Cu_{1.63}Se(1/3)$, and $Cu_2Se$, respectively.

The reduction of $CO_2$ to $CH_3OH$ could occur via a pathway involving six elementary steps[67] by Eqs. (5–10):

$$CO_2 + (H^+ + e^-) + * \rightarrow *COOH \quad (5)$$

$$*COOH + (H^+ + e^-) \rightarrow *CO + H_2O \quad (6)$$

$$*CO + (H^+ + e^-) \rightarrow *CHO \quad (7)$$

$$*CHO + (H^+ + e^-) \rightarrow *OCH_2 \quad (8)$$

$$*OCH_2 + (H^+ + e^-) \rightarrow *OCH_3 \quad (9)$$

$$*OCH_3 + (H^+ + e^-) \rightarrow CH_3OH + * \quad (10)$$

where * denotes the active sites on the catalyst surface. Based on the above mechanism, the free energies of the corresponding intermediate states are important to identify the activity of a given material in catalyzing $CO_2$ reduction. The computational hydrogen electrode (CHE) model[68] proposed by Norskov et al. was used to calculate the free energies of $CO_2$ reduction intermediates, based on which the free energy of an adsorbed species is defined as Eq. (11):

$$\Delta G_{ads} = \Delta E_{ads} + \Delta E_{ZPE} - T\Delta S_{ads} + \int C_P dT \quad (11)$$

where $\Delta E_{ads}$ is the electronic adsorption energy, $\Delta E_{ZPE}$ stands for the zero point energy difference between adsorbed and gaseous species, $T\Delta S_{ads}$ denotes the corresponding entropy difference between these two states, and $\int C_P dT$ is the enthalpy correction. The electronic binding energy is referenced as graphene for each C atom, ½ $H_2$ for each H atom, and ($H_2O - H_2$) for each O atom, plus the energy of the clean slab. The corrections of zero point energy, entropy, and enthalpy of adsorbed can be found in Supplementary Table 9.

**Code availability**. All code supporting the findings of this study are available from the corresponding author on request.

## Data availability

Data for Figs. 1 to 4, Supplementary Figures 1 to 34 and Supplementary Tables 1 to 9 can be found in the Source Data file.

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

## Acknowledgements

The work was supported by National Key Research and Development Program of China (2017YFA0403102), National Natural Science Foundation of China (21773267, 21733011, 21533011), Beijing Municipal Science & Technology Commission (Z181100004218004), and the Chinese Academy of Sciences (QYZDY-SSW-SLH013). The EXAFS experiment was conducted at Beijing Synchrotron Radiation Facility.

## Author contributions

D.X.Y., Q.G.Z., and B.X.H. proposed the project, designed the experiments, and wrote the manuscript; D.X.Y. performed the whole experiments; C.J.C., H.Z.L., Z.M.L., Z.J.Z., X.Y.Z., and S.J.L. performed the analysis of experimental data; B.X.H. and Q.G.Z. supervised the whole project.

## Additional information

**Competing interests:** The authors declare no competing interests.

