## [Peer Review File · Nature Communications]

Reviewers' comments:

Reviewer #1 (Remarks to the Author):

Yang et al produce a very interesting report on the use of copper selenide nanoparticles as electrode materials for the electrochemical conversion of carbon dioxide to methanol in aqueous ionic liquid solution. While the use of nanostructured transition metal chalcogenides in electrochemical CO₂ reduction has precedent (see refs 26 and 27, for example) the findings presented here are impactful as the use of copper selenides appears to bias product formation towards the 6e⁻, 6H⁺ reduction product methanol, of interest as a liquid energy carrier and fuel precursor. In comparison, previously reported metal selenide electrocatalysts showed high selectivity for carbon monoxide (ref 26). The catalysts are well characterized, and the products are supported by a variety of chemical analysis techniques (NMR, GC-MS).

In general, this is novel, exciting work and strong evidence is provided for its conclusions. I believe it will very important to researchers in the field moving forward, as engineering of binary Cu/Se particles may now be fine tuned to further improve catalyst performance to the point where it may be practical for fuel-forming purposes, and therefore interesting to researchers in related disciplines as well.

To help influence thinking in the field, the manuscript would be greatly strengthened, in my opinion, with some careful consideration of the potential mechanism and role of Se. Better referencing of work on metallic copper, known to mediate the reduction of CO₂ to gaseous hydrocarbons, in particular ethylene and methane (Hori, Chem. Lett. 1985, 11, 1695), and surface modified Cu (see for example, ACS Cent. Sci. 2017, 3, 853–859), would help bolster the impact of this work. Pending this revision, I would recommend acceptance for publication.

Other suggested revisions/corrections:

Line 23 – “store”

Line 40 – what is meant by “reasonable current density”? Can you give a range and a justification? This is an important point to make, given the justification of the importance of the findings presented here being the formation of MeOH with both high J and high Faradaic efficiency. On this note, a justification for the range given is also important, as to support the assertion that this set up may be useful for fuel forming purposes, it might be relevant to discuss what makes for a practically useful current density. I take similar issue with the idea of CuSe materials being described as “low-cost” (line 46).

Line 51 – I find the nomenclature “Cu_{2-x}Se_y... where y represents the VDETA/VH₂O” confusing, especially the use of a dash to represent both a minus sign (2-x) and a hyphen (-y) in the same breath. Is it possible to use an alternative nomenclature or to leave the solvent ratio out?

Line 90 – “formulae”

Line 104 – “AcN” – acetonitrile? “catholyte” – electrolyte? In general, I’m not sure this is the appropriate nomenclature. Please check. See also line 236.

Line 106 – with respect to CO₂ binding (later sulfate is tested as well, Figure 3b), this feels less important perhaps than the binding affinity of further products, as multi-electron, multi-proton steps are required to be orchestrated by the CuSe electrodes to manage to form methanol. Can the authors comment on CO binding affinity? See, for example, discussion in J. Am. Chem. Soc., 2014, 136 (40), pp 14107–14113. See also Figure 3, which suggests a mechanism that relies on strong binding of intermediates (such as CO) to the electrode surface to explain the selectivity for MeOH.

Line 116 – how was overpotential calculated? Please note which thermodynamic potential (with respect to pH) is being used? -0.38 V vs NHE for CO₂ to MeOH?

Line 118 – “replace”

Line 124 – can the authors comment on surface-bound stabilizing ligands on their nanoparticles? Does the amine persist on the surface of the NPs? Do they ripen over time?

Line 132 – I’m not sure I agree with the assertion that the Tafel slope reveals a fast 1e⁻ pre-

equilibrium step to form the CO₂ radical anion. The Tafel slope, for example, of the Sn/graphene catalysts mentioned in reference 40 (cited here) is just over half as large. Is the comparison here between various CuSe nanoparticle materials prepared here? Please clarify language if so.

Line 142 – “stabilizing”

Line 164 – sulfate seems like an odd choice. Firstly, it is dianionic, with a very different geometry to reduced CO₂ radical anion. Can the authors elaborate more on their choice of proxy adsorbent?

Line 171 – “proposed”

Line 177–Reference 43 does not support this assertion “... ionic liquid containing electrolyte and CO₂ formed a complex-[Bmim-CO₂]+”. Both modeling and spectroscopic studies (cited in reference 43) describe that with respect to solutions of CO₂ imidazolium ionic liquids, it is the anion that dominates the interactions with the CO₂, with the cation playing a secondary role. In fact, this is at odds slightly with reference 2, which employs a Ag cathode and sees CO as the dominant product. Could the authors comment on the role of the IL in light of reference 43? Could it simply be to enhance to local CO₂ concentration?

Line 177 – “electron”

Line 187 – This seems a missed opportunity to discuss in more depth the work in references 26 and 27, which indeed demonstrate that “other transition metal selenides” can be efficient electrocatalysts for CO₂ reduction, albeit with selectivity for CO rather than methanol.

Line 240 – “AcN-d6” do you mean CD₃CN?

Line 290 – incorrect journal abbreviation in reference

Reviewer #2 (Remarks to the Author):

In this manuscript, the authors reported the synthesis and characterization of copper selenide nanocatalysts for electrochemical reduction of CO₂ into methanol. They claim that the cooperative effect of Cu and Se accelerates the reaction by electrochemical study, including linear sweep voltammetry and electrochemical impedance spectroscopy. The presented copper selenide catalyst exhibited a high faradaic efficiency of over 76.2% at a low overpotential of 285 mV. The results are interesting in terms of synthesis and applications of copper selenide towards CO₂ reduction. Unfortunately, this manuscript lacks some key structural characterizations, and the theoretical investigation is not touched to clarify the mechanism. There are still some critical problems that should be clarified before this manuscript can be considered for publication. Some questions and suggestions are listed below.

1. The authors synthesized the nonstoichiometric Cu_{2-x}Se_y nanocatalysts and obtained a series of samples with different atomic ratio by controlling the value of VDETA/VH₂O. However, the structure of Cu_{2-x}Se_y is not clear. What is the influence of the ratio of VDETA/VH₂O on the Cu_{2-x}Se_y? They should provide sufficient critical structural characterizations (e.g., XPS, HRTEM, XAFS etc.) to confirm and discuss the relationship between the electrochemical performance and structure.
2. The authors claim that the Cu_{1.63}Se_{-1/3} catalyst has the higher binding affinity for CO₂•⁻ intermediates by examining the absorption of sulfate on different electrodes. They should supplement the theoretical calculations involving the binding affinity for CO₂•⁻ intermediates on different samples to verify the experimental results.
3. In Figure 4, the authors propose the reaction mechanism for electrocatalytic CO₂ reduction to methanol on Cu_{2-x}Se_y electrode. However, the experimental data and results are not enough to support the author's statement. Additional experimental measurements are needed to further determine the reaction pathway, such as IR spectroscopy study during the reaction to observe the CO₂•⁻ intermediates directly.
4. The author should check the typos and grammar more carefully. For example: (1) Line 118 on

page 7, the text "relace" should be "replace". (2) Line 171 on page 10, the text "Prpposed" should be "Proposed". (3) Line 171 on page 10, the annotation "Figure 3" should be "Figure 4".

Reviewer #3 (Remarks to the Author):

The manuscript by Han and co-workers describes Cu selenide electrodes for CO₂ reduction in electrolytes composed of ionic liquid, acetonitrile, and water. The work appears technically sound, but the significance of the results is not high enough to warrant publication in Nat. Commun. In addition, some of the conclusions are not fully supported by the data. In particular:

1) The need to operate in BmimPF₆/ACN/H₂O electrolyte compromises the utility of the catalyst. The overpotential at the cathode may be low, but it is not clear how the electrolysis could be performed with a low cell potential. The cell potential is determined by the electrode overpotentials and the voltage required for ion transport (*iR*). The anode is operating in acid, which will require >+1.2 V vs Ag/AgCl. Combined with the -2 V at the cathode, and an unspecified but likely large *iR*, the overall cell voltage will be ~3.5 V, which makes the process very inefficient. In addition, separating methanol from the electrolyte would be very challenging without a large energy input. Optimizing catalysis under conditions that are not amenable to practical electrosynthesis is perhaps fundamentally interesting but, in my opinion, not suitable for this journal.

2) The comparisons in Fig.3a and S14 do not take into account surface area. It is very difficult to assess differences in activity across materials when the electrode morphologies/surface areas are significantly different.

3) The LSVs in Fig. 3b are used to probe "surface adsorption" of sulfate. It is not clear how to interpret the data. What does the current correspond to? These are not adsorption features and represent either oxidation of the electrode itself or water. In any case, the adsorption of sulfate under strong anodic potential has nothing to do with the adsorption of CO₂ reduction intermediates.

4) The authors propose hydrogen bonding between CO₂ and the Bmim⁺ electrolyte. CO₂ is a terrible hydrogen bond acceptor and bmim is a rather weak donor. What is the evidence for this interaction?

Responses to the comments and revisions made

Referee 1

Remarks to the Author

Yang et al produce a very interesting report on the use of copper selenide nanoparticles as electrode materials for the electrochemical conversion of carbon dioxide to methanol in aqueous ionic liquid solution. While the use of nanostructured transition metal chalcogenides in electrochemical CO₂ reduction has precedent (see refs 26 and 27, for example) the findings presented here are impactful as the use of copper selenides appears to bias product formation towards the 6e⁻, 6H⁺ reduction product methanol, of interest as a liquid energy carrier and fuel precursor. In comparison, previously reported metal selenide electrocatalysts showed high selectivity for carbon monoxide (ref 26). The catalysts are well characterized, and the products are supported by a variety of chemical analysis techniques (NMR, GC-MS).

In general, this is novel, exciting work and strong evidence is provided for its conclusions. I believe it will very important to researchers in the field moving forward, as engineering of binary Cu/Se particles may now be fine tuned to further improve catalyst performance to the point where it may be practical for fuel-forming purposes, and therefore interesting to researchers in related disciplines as well.

To help influence thinking in the field, the manuscript would be greatly strengthened, in my opinion, with some careful consideration of the potential mechanism and role of Se. Better referencing of work on metallic copper, known to mediate the reduction of CO₂ to gaseous hydrocarbons, in particular ethylene and methane (Hori, Chem. Lett. 1985, 11, 1695), and surface modified Cu (see for example, ACS Cent. Sci. 2017, 3, 853–859), would help bolster the impact of this work. Pending this revision, I would recommend acceptance for publication.

Response: We thank the referee for the comment. On the basis of the comment, we have made the following modifications.

1) To explain the role of Se in the catalysts, the experimental results for electroreduction CO₂ to methanol over various catalysts, including Cu, CuO, Cu₂O,

CuS, Cu₂S, CuSe, Cu₂Se and Cu_{1.63}Se(1/3), have been provided in Fig. 3a. It was found that Cu_xSe catalysts had better performance for CO₂ reduction to methanol. We can deduce that Se in the catalysts is crucial for efficient CO₂ reduction to methanol. In the revised manuscript, we have discussed this by “Thus, on the basis of above results, we can deduce that Se in the catalysts is crucial for efficient CO₂ reduction to methanol.” (Please see Page 10 of the revised manuscript).

2) To explain the role of Se in the catalyst, the EXAFS experiments were also conducted to characterize the catalysts. We have discussed this by adding “We also carried out extended X-ray absorption fine structure spectroscopy (EXAFS) experiments to study Cu K-edge, which can disclose the local atomic arrangements of the catalysts. The Cu K-edge $k^2\chi(k)$ oscillation curve for Cu_{1.63}Se(1/3) was obviously different from that for CuSe and Cu₂Se (Supplementary Fig. 25). It can be seen that the disorder degree in Cu_{1.63}Se(1/3) was remarkably larger and the bond length of Cu-Se in Cu_{1.63}Se(1/3) was between that in CuSe and Cu₂Se (Supplementary Figs. 26-29 and Supplementary Table 6). All these results indicate the presence of a noticeable structure distortion in Cu_{1.63}Se(1/3) catalyst, which may affect the performance for CO₂ electroreduction.” (Please see Page 10 of the revised manuscript).

To verify the influence of the Cu_{1.63}Se(1/3) intrinsic structure on CO₂ electroreduction, we have carried out density functional theory (DFT) study for the reaction over Cu_{1.63}Se(1/3), Cu₂Se and CuSe. In the revised manuscript, we have discussed this by “The density functional theory (DFT) calculations were also conducted on the multiple elementary reaction steps, and the results are shown in Fig. 4b and Supplementary Figs. 30-32. Comparing with other two catalysts (Cu₂Se and CuSe), the formation of intermediate (*COOH) on the Cu_{1.63}Se(1/3) surfaces can reach a stable configuration with lower free energy via two neighboring Cu atoms through Cu-C and Cu-O bonds. The *COOH intermediate bind with the active sites on the surface of catalysts and accelerate the formation of adsorbed *CO species⁵⁴. The Cu_{1.63}Se(1/3) catalyst also has a moderate binding energy for *CO among the

three catalysts, which is beneficial for CO₂ transformation to more reduced products that require more than a two-electron reduction^{20, 53} It can also be seen that the step of *CO reduction to *CHO was an endothermic and rate-limiting process. Compared with Cu₂Se and CuSe, the free energy of *CHO over Cu_{1.63}Se(1/3) catalyst is more negative, which may be mainly originated from the moderately strong binding energy for *CO intermediate. In addition, the C-Cu bond (Supplementary Fig. 33) between Cu_{1.63}Se(1/3)-CHO is 1.926 Å, which is shorter than those of Cu₂Se-CHO (2.188 Å) and CuSe-CHO (2.002 Å), indicating that *CHO is easier to adsorb on the surface of the catalyst to accept electrons and protons to form *OCH₂ and *OCH₃, and then is reduced to methanol. These results illustrate that the structure distortion of Cu_{1.63}Se(1/3) was beneficial for CO₂ electroreduction to methanol.” (Please see Page 11 of the revised manuscript).

3) The suggested references (Chem. Lett. 1985, 11, 1695 and ACS Cent. Sci. 2017, 3, 853–859) have also been discussed in the revised manuscript by “Metal and metal-based catalysts have been used for CO₂ electroreduction to CO, hydrocarbons and alcohols^{15, 16}.”. Please see page 2 in the revised manuscript.

Other suggested revisions/corrections:

1. Line 23- “store”

Response: We thank the referee for the comment. The “store” has been changed to “transform” in the revised manuscript.

2. Line 40-what is meant by “reasonable current density”? Can you give a range and a justification? This is an important point to make, given the justification of the importance of the findings presented here being the formation of MeOH with both high J and high Faradaic efficiency. On this note, a justification for the range given is also important, as to support the assertion that this set up may be useful for fuel forming purposes, it might be relevant to discuss what makes for a practically useful current density. I take similar issue with the idea of CuSe materials being described as

“low-cost” (line 46).

Response: We thank the referee for the comment. We agree with the referee that the meaning of “reasonable current density” is not clear. In the revised manuscript, we have changed “Nevertheless, achieving high current density and FE simultaneously for conversion of CO₂ to methanol remains to be a challenge, and only a few catalysts reported up to date could reach high selectivity with reasonable current density” in the original manuscript into “Nevertheless, achieving high current density and FE simultaneously for conversion of CO₂ to methanol remains to be a challenge, and only a few catalysts reported up to date could reach relatively high current density (> 10 mA cm⁻²) and selectivity (> 50%), as shown in Supplementary Table 1. Therefore, designing efficient catalysts to enhance the activity and FE, and reduce the overpotential is very interesting from both scientific and practical viewpoints.” (Please see Page 2 of the revised manuscript). We also have given a table to list the reported results for electrochemical reduction of CO₂ to methanol (Supplementary Table 1). As for the low cost, we got this argument from the review paper (Ref. 25). We agree with the referee that “low-cost” is confusing without comparing with other materials. Thus, according to the comment, we have also changed “They are also low-cost materials that have structure stability and composition-dependent optical/electrical properties^{25,33}.” into “They also have structure stability and composition-dependent optical/electrical properties³³. In addition, they are also low-cost materials comparing with many other materials²⁵, especially noble metals.” (Please see Page 3 in the revised manuscript)

3. Line 51-I find the nomenclature “Cu_{2-x}Se-y” where y represents the V_{DETA}/V_{H₂O}” confusing, especially the use of a dash to represent both a minus sign (2-x) and a hyphen (-y) in the same breath. Is it possible to use an alternative nomenclature or to leave the solvent ratio out?

Response: We thank the referee for the comment. As suggested by the reviewer, in the revised manuscript, we have changed the nomenclature “Cu_{2-x}Se-y” to

“Cu_{2-x}Se(y)”.

4. Line 90- “formulae”

Response: We thank the referee very much for the comment. According to the suggestion of referee, “formulas” has been modified to “formulae”. (Please see Page 5 in the revised manuscript)

5. Line 104- “AcN” -acetonitrile? “catholyte” -electrolyte? In general, I’m not sure this is the appropriate nomenclature. Please check. See also line 236.

Response: We thank the referee for the comment. In this manuscript, “AcN” stands for acetonitrile. To avoid confusion, we have changed “AcN” into “CH₃CN” and “catholyte” into “electrolyte”.

6. Line 106-with respect to CO₂ binding (later sulfate is tested as well, Figure 3b), this feels less important perhaps than the binding affinity of further products, as multi-electron, multi-proton steps are required to be orchestrated by the CuSe electrodes to manage to form methanol. Can the authors comment on CO binding affinity? See, for example, discussion in J. Am. Chem. Soc., 2014, 136 (40), pp 14107-14113. See also Figure 4, which suggests a mechanism that relies on strong binding of intermediates (such as CO) to the electrode surface to explain the selectivity for MeOH.

Response: We thank the referee for the comment. We agree with the referee that the binding affinity of CO is important for producing further products. Our control experiments indicate that CO clearly promoted the formation of methanol and thus it is a possible intermediate in the reaction. This is also consistent with result of our density functional theory (DFT). Previous reports also indicate that product yield and selectivity of the CO₂ reduction reaction depend on binding energy of CO and catalysts. Catalyst that binds CO strongly produces few CO₂ reduction reaction products, whilst that with low binding energy produces mostly CO. Therefore, the moderate CO binding affinity on Cu selenide is crucial to enhancing the selectivity of

methanol. However, Cu based catalyst possesses an intermediate binding energy for CO, which is believed to be the reason for its ability to catalyze the formation of more reduced products^{20, 53}. In the revised manuscript, we have studied the *CO binding affinity of Cu_{1.63}Se(1/3), Cu₂Se and CuSe by DFT calculation. Result indicates that Cu_{1.63}Se(1/3) catalyst has a moderately binding energy for *CO among the three catalysts. The moderately strong binding energy for *CO can be advantageous for forming *CHO, which is a vital intermediate for further transformation to more reduced products that require more electrons. In the revised manuscript, we have discussed this by “The Cu_{1.63}Se(1/3) catalyst also has a moderate binding energy for *CO among the three catalysts, which is beneficial for CO₂ transformation to more reduced products that require more than a two-electron reduction^{20, 53} It can also be seen that the step of *CO reduction to *CHO was an endothermic and rate-limiting process. Compared with Cu₂Se and CuSe, the free energy of *CHO over Cu_{1.63}Se(1/3) catalyst is more negative, which may be mainly originated from the moderately strong binding energy for *CO intermediate.” Please see Page 11 in the revised manuscript.

7. Line 116-how was overpotential calculated? Please note which thermodynamic potential (with respect to pH) is being used? -0.38 V vs NHE for CO₂ to MeOH?

Response: We thank the referee for the comment. The main method to get thermodynamic potential (equilibrium potential) is theoretical calculation using thermodynamic parameters at the experimental condition. $E^0_{\text{CO}_2/\text{CH}_3\text{OH}}$ is known as a function of the pH when aqueous electrolyte is used (Ref, Bard, A. J., Parsons, R. & Jordan, J., Ed., Standard Potentials in Aqueous Solution (Marcel Dekker, New York, 1995).). Adaptation to ILs and acetonitrile, the electron, acidity, and CO₂-stoichiometry should be taken into account to estimate the thermodynamic potential (Ref, Costentin, C. *et al. Science* **338**, 90-94, (2012).). However, we could not find the thermodynamic potential for CO₂ to CH₃OH in IL system after thorough literature survey because most of the parameters to estimate the $E^0_{\text{CO}_2/\text{CH}_3\text{OH}}$ are not known. Therefore, we used the extrapolation method, which is also commonly used⁴³⁻⁴⁵ to obtain the thermodynamic potential of this reaction. In this method, the

thermodynamic potential is obtained by extrapolation of partial current density vs potential curve to zero partial current density. As suggested by the referee, we have given a more detailed description on the overpotential. We have emphasized this by “The equilibrium (thermodynamic) potential for CH₃OH was -1.815 V vs Ag/Ag⁺, which was obtained by extrapolation of partial current density vs potential curve to zero partial current density (Supplementary Fig. 11)⁴³⁻⁴⁵.” Please see page 7 in the revised manuscript.

At the same time, in the Supplementary Information, the detailed method has been discussed by “Overpotential (η) is the difference between the equilibrium potential and the actual potential for the transformation of the substrate CO₂ into the product methanol: $\eta = E - E^0_{\text{CO}_2 \rightarrow \text{methanol}}$. Here, the $E^0_{\text{CO}_2 \rightarrow \text{methanol}}$ referred to the equilibrium potential for CO₂ transformation to CH₃OH, which can be obtained by extrapolation method⁴³⁻⁴⁵. Taking the Cu_{1.63}Se(1/3) electrode as example, stepped potential electrolysis experiments between -1.8 V and -2.1 V were carried out and the electrolysis products were collected and characterized. The current densities for CH₃OH at each potential are shown in Supplementary Fig. 11, and the potential at $j_{\text{CH}_3\text{OH}}=0$ by extrapolation method is the equilibrium potential. Therefore, the overpotential can be obtained. The method to calculate the overpotential over other electrodes was similar.”. (Please see page 2 in Supplementary Information).

8. Line 118- “replace”

Response: We thank the referee for the comment. The wrong spelling has been corrected.

9. Line 124-can the authors comment on surface-bound stabilizing ligands on their nanoparticles? Does the amine persist on the surface of the NPs? Do they ripen over time?

Response: We thank the referee for the comment. According to the XPS results before and after electrolysis (Supplementary Fig. 15), it can be known that no amine persists in the catalysts. After reading the comment, we also carried out the TG

analysis to further verify this. TG curve is shown in Supplementary Fig. 3. The weight loss of DETA (if existed) should occur before 280 °C-580 °C³⁶. It can be known from the figure that there was no weight loss in the temperature range, indicating that there was no DETA in the catalysts. We have described this by “TG curve is shown in Supplementary Fig. 3. The weight loss of DETA (if existed) should occur at 280 °C-580 °C³⁶. It can be seen from the figure that there was no weight loss in the temperature range, indicating that there was no DETA in the catalysts.” in the revised manuscript. Please see page 5 in the revised manuscript.

10. Line 132-I’m not sure I agree with the assertion that the Tafel slope reveals a fast $1e^-$ pre-equilibrium step to form the CO_2 radical anion. The Tafel slope, for example, of the Sn/graphene catalysts mentioned in reference 40 (cited here) is just over half as large. Is the comparison here between various CuSe nanoparticle materials prepared here? Please clarify language if so.

Response: We thank the referee for the comment. As suggested by the referee, we have modified the explanation with clarified language by “The Tafel slope of $Cu_{1.63}Se(1/3)$ was smaller than other $Cu_{2-x}Se(y)$ catalysts, which leads to faster increment of CO_2 reduction rate with increasing overpotential^{26,47}.” Please see page 7 in the revised manuscript.

11. Line 142- “stabilizing”

Response: We thank the referee for the comment. The wrong spelling has been corrected.

12. Line 164-sulfate seems like an odd choice. Firstly, it is dianionic, with a very different geometry to reduced CO_2 radical anion. Can the authors elaborate more on their choice of proxy adsorbent?

Response: We thank the referee for the comment. We agree with the referee that the property of sulfate is different from that of reduced CO_2 radical anion and the adsorption of sulfate is not equal to the adsorption of reduced CO_2 radical anion

quantitatively. However, sulfate is often used as the analogs of the CO_2^- ion to qualitatively compare the variation of binding energy for CO_2^- intermediates on various materials^{21, 49, 51} because it is very difficult to find better analogs considering the various factors, such as chemical stability at the electrochemical reaction condition. Therefore, we used sulfate in this work, and the results support our conclusion. Besides sulfate, we also tried to use both formate and acetate, but they are not stable at the experimental condition. However, this is only one of experimental evidences to explain why $\text{Cu}_{1.63}\text{Se}(1/3)$ showed better performance. If the referee insists on removing this, we would like to do this because the conclusion of the paper is the same without these results. In revised manuscript, we have discussed this as following.

“It is known that smaller overpotential of hydroxyls and sulfate adsorption indicates larger binding energy of intermediates⁵¹. Therefore, hydroxyls and sulfates (or bisulfates) can be used as the analogs of the CO_2^- ion to study the binding energy of intermediates. In this work, we studied the binding affinity for CO_2^- intermediates on various electrodes using sulfate, and the method is similar to that in the previous reports^{21, 49, 51}. Fig. 3b shows LSVs of the sulfate adsorption peaks on different catalysts in 0.1 M sulfuric acid. It can be known that the sulfate binds more strongly on $\text{Cu}_{1.63}\text{Se}(1/3)$.” Please see page 9 in the revised manuscript.

13. Line 171- “proposed”

Response: We thank the referee for the comment. The wrong spelling has been corrected.

14. Line 177-Reference 43 does not support this assertion “... ionic liquid containing electrolyte and CO_2 formed a complex- $[\text{Bmim-CO}_2]^+$ ”. Both modeling and spectroscopic studies (cited in reference 43) describe that with respect to solutions of CO_2 imidazolium ionic liquids, it is the anion that dominates the interactions with the CO_2 , with the cation playing a secondary role. In fact, this is at odds slightly with reference 2, which employs a Ag cathode and sees CO as the dominant product.

Could the authors comment on the role of the IL in light of reference 43? Could it simply be to enhance to local CO₂ concentration?

Response: We thank the referee for the comment. In this work, we used imidazolium based ILs electrolytes for several reasons. First, imidazolium based ILs combine the advantages of large conductivity, wide electrochemical window, wider liquid range, and high stability (Ref, Galiński M., Lewandowski A. & Stępnik I. *Electrochimica acta* **51**, 5567-5580, (2006).). Second, CO₂ is more soluble in imidazolium based ILs⁵². Third, CO₂ can form complex with the imidazolium cations, which can enhance the CO₂ concentration in electrolyte and transport CO₂ to the catalyst surface to improve further reduction of CO₂².

We also agree with the referee that anion in the ionic liquid also plays an important role for interaction between CO₂ and ionic liquids⁵², which in turn influences the electrochemical reaction. After reading the comment, we carried out some new experiments using other commonly used ionic liquids with different anions, such as [Bmim]PF₆, [Bmim]BF₄, [Bmim]TF₂N, [Bmim]OAc, [Bmim]NO₃, and [Bmim]ClO₄, and acetonitrile system for comparison. The data are given in Supplementary Figs. 21 and 22. It can be found that the current density for CO₂ reduction could be enhanced when the ionic liquids were added in the electrolytes. The results demonstrated that the anions also had a great influence on the catalytic activity of CO₂ reduction. From the results in Supplementary Figs. 21 and 22, it can be known that [Bmim]PF₆ exhibited much higher current density and Faradaic efficiency for methanol, further indicating that anions played important role in the reaction, which resulted partially from the difference of the interaction between CO₂ and the ILs with different anions. In the revised manuscript, we have discussed this by “To further understand the role of anions in the ILs, other ILs were also used, including [Bmim]PF₆, [Bmim]BF₄, [Bmim]TF₂N, [Bmim]OAc, [Bmim]NO₃, and [Bmim]ClO₄, and acetonitrile systems for comparison (Supplementary Figs. 21 and 22). It can be observed that the anions of the ILs also influenced the electrochemical reaction significantly (Supplementary Fig. 22), which resulted partially from the difference of the interaction between CO₂ and the anions of the ILs. [Bmim]PF₆

exhibited higher current density and Faradaic efficiency for methanol among all the ILs used.”. Please see page 8 in the revised manuscript.

Ionic liquids can not only enhance local CO₂ concentration by complexation, but also influence the catalytic activity of CO₂ reduction. The statement in the original manuscript has been modified as “The electrolyte containing ionic liquids and CO₂ formed complex CO₂-[Bmim]PF₆, which can enhance the CO₂ concentration in electrolyte and transport CO₂ to the catalyst surface to improve further reduction of CO₂.²” Please see page 11 in the revised manuscript.

15. Line 177- “electron”

Response: We thank the referee very much for the comment. The wrong spelling has been corrected.

16. Line 187-This seems a missed opportunity to discuss in more depth the work in references 26 and 27, which indeed demonstrate that “other transition metal selenides” can be efficient electrocatalysts for CO₂ reduction, albeit with selectivity for CO rather than methanol.

Response: We thank the referee for the comment. Some discussions have been added in the revised manuscript by “WSe₂ nanoflakes was reported as an efficient catalyst for CO₂ electroreduction to CO²⁹. Density functional theory (DFT) calculation indicated that molybdenum sulfides and selenides were also possible catalysts for CO₂ electroreduction³⁰, which showed that the intermediates COOH and CHO were more easily adsorbed on the S and Se atoms at the edges than the intermediate CO. Therefore, transition-metal selenides may be a class of promising catalysts for CO₂ electroreduction.” Please see page 3 in the revised manuscript.

17. Line 240- “AcN-d₆” do you mean CD₃CN?

Response: We thank the referee very much for the comment. The meaning of “AcN-d₆” was CD₃CN, and “AcN-d₆” has been changed to “CD₃CN” in the revised manuscript.

18. Line 290-incorrect journal abbreviation in reference

Response: We thank the referee very much for the comment. The journal abbreviation has been corrected.

Referee 2

Remarks to the Author

In this manuscript, the authors reported the synthesis and characterization of copper selenide nanocatalysts for electrochemical reduction of CO₂ into methanol. They claim that the cooperative effect of Cu and Se accelerates the reaction by electrochemical study, including linear sweep voltammetry and electrochemical impedance spectroscopy. The presented copper selenide catalyst exhibited a high faradaic efficiency of over 76.2% at a low overpotential of 285 mV. The results are interesting in terms of synthesis and applications of copper selenide towards CO₂ reduction. Unfortunately, this manuscript lacks some key structural characterizations, and the theoretical investigation is not touched to clarify the mechanism. There are still some critical problems that should be clarified before this manuscript can be considered for publication. Some questions and suggestions are listed below.

1. The authors synthesized the nonstoichiometric Cu_{2-x}Se-y nanocatalysts and obtained a series of samples with different atomic ratio by controlling the value of V_{DETA}/V_{H₂O}. However, the structure of Cu_{2-x}Se-y is not clear. What is the influence of the ratio of V_{DETA}/V_{H₂O} on the Cu_{2-x}Se-y? They should provide sufficient critical structural characterizations (e.g., XPS, HRTEM, XAFS etc.) to confirm and discuss the relationship between the electrochemical performance and structure.

Response: We thank the referee for the comment. In this work, the Cu_{2-x}Se(y) catalysts were synthesized by solvent coordination molecular template method^{34, 35}, which is shown schematically in Supplementary Fig. 1. The protonated amine can act as template molecules for forming the structured nanocrystals. That is, the size and morphology of Cu_{2-x}Se(y) catalysts were tuned by the volume ratios of DETA and water (V_{DETA}/V_{H₂O}) in the mixed solvent.

After reading the comment, we carried out more experiments to characterize the catalysts using XPS, HRTEM and EXAFS, which are discussed in the following.

The XPS and HRTEM results of the catalysts are given in Supplementary Figs. 4-8. We have discussed this by “The results showed that the crystal structures of different samples were not changed with the $V_{\text{DETA}}/V_{\text{H}_2\text{O}}$ ratio notably. However, the size and morphology depended strongly on the composition of the solvents, which influenced the performances of CO_2 electroreduction.”. Please see page 5 in the revised manuscript.

We have also conducted EXAFS experiments, which can show the local atomic arrangements of the catalysts, and the results are given in Supplementary Figs. 25-29 and Supplementary Table 6. We have discussed this by “We also carried out extended X-ray absorption fine structure spectroscopy (EXAFS) experiment to study Cu K-edge, which can disclose the local atomic arrangements of the catalysts. The Cu K-edge $k^2\chi(k)$ oscillation curve for $\text{Cu}_{1.63}\text{Se}(1/3)$ was obviously different from that for CuSe and Cu_2Se (Supplementary Fig. 25). It can be seen that the disorder degree in $\text{Cu}_{1.63}\text{Se}(1/3)$ was remarkably larger and the bond length of Cu-Se in $\text{Cu}_{1.63}\text{Se}(1/3)$ was between that in CuSe and Cu_2Se (Supplementary Figs. 26-29 and Supplementary Table 6). All these results indicate the presence of a noticeable structure distortion in $\text{Cu}_{1.63}\text{Se}(1/3)$ catalyst, which may affect the performance for CO_2 electroreduction.”. Please see page 10 in the revised manuscript.

We have also given the EXAFS experimental details in the revised manuscript by “The homogeneously mixed samples (20 mg) and graphite (100 mg) sample were pressed into circular slices with a diameter of 10 mm which was used for further EXAFS measurement under ambient condition. The data of EXAFS were obtained via the beamline 1W1B station of BSRF (Beijing Synchrotron Radiation Facility, P.R. China). The storage rings of BSRF were operated at 2.5 GeV with the maximum current of 250 mA. Si (111) double-crystal monochromator crystals were used to monochromatize the X-ray beam and the detuning was done by 10% to remove harmonics. EXAFS data were collected in transmission mode in the energy range from -200 below to 1000 eV above the Cu K-edge. The acquired EXAFS data were processed according to the standard procedures using the ATHENA module implemented in the IFEFFIT software packages. The quantitative curve-fittings were carried out in the R-space with a Fourier transform k-space range of 2.2-12.8 \AA^{-1} using the module ARTEMIS of IFEFFIT. The backscattering amplitude $F(k)$ and

phase shift $\Phi(k)$ were calculated using FEFF 8.0 code.”. We believe that after adding the information, the experimental set-up and procedures are clear. (Please see Page 2 in the Supplementary Information).

2. The authors claim that the $\text{Cu}_{1.63}\text{Se}$ -1/3 catalyst has the higher binding affinity for $\text{CO}_2\bullet^-$ intermediates by examining the absorption of sulfate on different electrodes. They should supplement the theoretical calculations involving the binding affinity for $\text{CO}_2\bullet^-$ intermediates on different samples to verify the experimental results.

Response: We thank the referee for the comment. On the basis of the comment, we have conducted theoretical calculations, and the results support our conclusions. In the revised manuscript, we have discussed this by “The density functional theory (DFT) calculations were also conducted on the multiple elementary reaction steps, and the results are shown in Fig. 4b and Supplementary Figs. 30-32. Comparing with other two catalysts (Cu_2Se and CuSe), the formation of intermediate (*COOH) on the $\text{Cu}_{1.63}\text{Se}(1/3)$ surfaces can reach a stable configuration with lower free energy via two neighboring Cu atoms through Cu-C and Cu-O bonds. The *COOH intermediate binds with the active sites on the surface of catalysts and accelerate the formation of adsorbed *CO species⁵⁴. The $\text{Cu}_{1.63}\text{Se}(1/3)$ catalyst also has a moderate binding energy for *CO among the three catalysts, which is beneficial for CO_2 transformation to more reduced products that require more than a two-electron reduction^{20, 53}. It can also be seen that the step of *CO reduction to *CHO was an endothermic and rate-limiting process. Compared with Cu_2Se and CuSe , the free energy of *CHO over $\text{Cu}_{1.63}\text{Se}(1/3)$ catalyst is more negative, which may be mainly originated from the moderately strong binding energy for *CO intermediate. In addition, the C-Cu bond (Supplementary Fig. 33) between $\text{Cu}_{1.63}\text{Se}(1/3)\text{-CHO}$ is 1.926 Å, which is shorter than those of $\text{Cu}_2\text{Se-CHO}$ (2.188 Å) and CuSe-CHO (2.002 Å), indicating that *CHO is easier to adsorb on the surface of the catalyst to accept electrons and protons to form *OCH_2 and *OCH_3 , and then is reduced to methanol. These results illustrate that the structure distortion of $\text{Cu}_{1.63}\text{Se}(1/3)$ was beneficial for CO_2 electroreduction to

methanol.” Please see page 11 in the revised manuscript.

3. In Figure 4, the authors propose the reaction mechanism for electrocatalytic CO₂ reduction to methanol on Cu_{2-x}Se-y electrode. However, the experimental data and results are not enough to support the author's statement. Additional experimental measurements are needed to further determine the reaction pathway, such as IR spectroscopy study during the reaction to observe the CO₂•⁻ intermediates directly.

Response: We thank the referee for the comment. We agree with the referee that in-situ IR spectroscopy is a very useful technique to study the intermediate during a reaction. However, we cannot carry out the experiment because the composite electrode which include catalyst and carbon black, has strong adsorption of IR light. We also found researches in the literatures are concentrated on using pure metal electrode for in situ IR in the reaction (Ref, Liu, Y. M. *et al. J. Am. Chem. Soc.* **137**, 11631-11636, (2015); Baruch, M. F. *et al. ACS Catal.* **5**, 3148-3156, (2015); Ye, L. T. *et al. Nat. Commun.* **8**, 14785, (2017).). Therefore, we regret that we are unable to get this result since the electrochemical system is not up to the experimental conditions and it needs more special electrochemical cell. Thus, according to the comment, we tried our best to get the evidences to study the reaction pathway, which are discussed in the following.

1) We studied the composition of the electrolyte samples taking at different electrolysis times by FT-IR, and the results are given in Supplementary Fig. 13. The intensity of the characteristic absorption bands that appeared at 1085 cm⁻¹, 1140 cm⁻¹ and 1420 cm⁻¹, assigning to C-O (a), C-H (b) and CH₃ (c) increased with increasing electrolysis time. This phenomena indicated formation of methanol.

2) Some control experiments were conducted in the presence of the possible reaction intermediates, such as formic acid, CO and formaldehyde (Supplementary Table 7). From the production rates of methanol, it can be seen that CO and formaldehyde clearly promoted the formation of methanol, and thus they are reasonable intermediates in the formation of methanol. We have discussed this by “To

understand the reaction pathway for the formation of methanol, some control experiments were conducted in the presence of the possible reaction intermediates, such as formic acid, CO and formaldehyde (Supplementary Table 7). From the production rates of methanol, it can be seen that CO and formaldehyde clearly promoted the formation of methanol and thus they are possible intermediates in the formation of methanol.”. Please see page 11 in the revised manuscript.

3) To get insight into the CO₂ reduction reaction at the microscopic level, we carried out density functional theory (DFT) to study the mechanism. Theoretical calculation results were demonstrated in Fig. 4b and Supplementary Figs. 30-33 to compare the free energy of reaction intermediates during CO₂ reduction on Cu_{1.63}Se(1/3), Cu₂Se and CuSe, which further support the reaction mechanism for electrocatalytic CO₂ reduction to methanol on the electrodes. We have discussed the mechanism by adding “The density functional theory (DFT) calculations were also conducted on the multiple elementary reaction steps, and the results are shown in Fig. 4b and Supplementary Figs. 30-32. Comparing with other two catalysts (Cu₂Se and CuSe), the formation of intermediate (*COOH) on the Cu_{1.63}Se(1/3) surfaces can reach a stable configuration with lower free energy via two neighboring Cu atoms through Cu-C and Cu-O bonds. The *COOH intermediate bind with the active sites on the surface of catalysts and accelerate the formation of adsorbed *CO species⁵⁴. The Cu_{1.63}Se(1/3) catalyst also has a moderate binding energy for *CO among the three catalysts, which is beneficial for CO₂ transformation to more reduced products that require more than a two-electron reduction^{20, 53} It can also be seen that the step of *CO reduction to *CHO was an endothermic and rate-limiting process. Compared with Cu₂Se and CuSe, the free energy of *CHO over Cu_{1.63}Se(1/3) catalyst is more negative, which may be mainly originated from the moderately strong binding energy for *CO intermediate. In addition, the C-Cu bond (Supplementary Fig. 33) between Cu_{1.63}Se(1/3)-CHO is 1.926 Å, which is shorter than those of Cu₂Se-CHO (2.188 Å) and CuSe-CHO (2.002 Å), indicating that *CHO is easier to adsorb on the surface of the catalyst to accept electrons and protons to form *OCH₂ and *OCH₃, and then is reduced to methanol. These results illustrate that the structure distortion of

Cu_{1.63}Se(1/3) was beneficial for CO₂ electroreduction to methanol.” Please see page 11 in the revised manuscript.

4. The author should check the typos and grammar more carefully. For example: (1) Line 118 on page 7, the text “relace” should be “replace” . (2) Line 171 on page 10, the text “Prpposed” should be “Proposed” . (3) Line 171 on page 10, the annotation “Figure 3” should be “Figure 4” .

Response: We thank the referee for the comment. The wrong typos and grammar have been modified in the revised manuscript.

Referee 3

Remarks to the Author

The manuscript by Han and co-workers describes Cu selenide electrodes for CO₂ reduction in electrolytes composed of ionic liquid, acetonitrile, and water. The work appears technically sound, but the significance of the results is not high enough to warrant publication in Nat. Commun. In addition, some of the conclusions are not fully supported by the data. In particular:

1. The need to operate in BmimPF₆/AcN/H₂O electrolyte compromises the utility of the catalyst. The overpotential at the cathode may be low, but it is not clear how the electrolysis could be performed with a low cell potential. The cell potential is determined by the electrode overpotentials and the voltage required for ion transport (*iR*). The anode is operating in acid, which will require >+1.2 V vs Ag/AgCl. Combined with the - 2 V at the cathode, and an unspecified but likely large *iR*, the overall cell voltage will be ~3.5 V, which makes the process very inefficient. In addition, separating methanol from the electrolyte would be very challenging without a large energy input. Optimizing catalysis under conditions that are not amenable to practical electrosynthesis is perhaps fundamentally interesting but, in my opinion, not suitable for this journal.

Response: We thank the referee for the comment. The two questions are discussed separately below.

1) In this study, we carried the experiments using commonly used H-type electrolysis cell. We agree with the referee that the cell potential is determined by the electrode overpotentials and the voltage required for ion transport (iR) in an electrochemical reaction system, and we cannot change the theoretical voltage for the reaction. So researches in the literature are concentrated on increasing current density and Faradaic efficiency for CO_2 reduction on the cathode, and reducing the overpotentials. Here, we report the first work for electrochemical reduction of CO_2 using copper selenide as the catalyst. It was discovered that the $\text{Cu}_{1.63}\text{Se}(1/3)$ nanocatalysts yielded outstanding current density of 41.5 mA cm^{-2} with FE of 77.6% at $-2.1 \text{ V vs. Ag/Ag}^+$. The current density is much higher than those reported up to date with very high FE for producing methanol (Supplementary Table 1) and the overpotential is low (285 mV). Thus, we believe that the work meets the high standard of the journal.

2) We also agree with the referee that separation of a polar product is a very important factor for practical application. In this work, we mainly report our discovery that $\text{Cu}_{1.63}\text{Se}(1/3)$ nanocatalysts is an outstanding electrocatalyst as discussed above, and the separation of methanol is not studied. Here we would like to discuss the separation briefly in response to the comment. We used $[\text{Bmim}]\text{PF}_6/\text{CH}_3\text{CN}/\text{H}_2\text{O}$ ternary electrolyte. $[\text{Bmim}]\text{PF}_6$ is not volatile, while CH_3OH , CH_3CN and H_2O are volatile. Thus, the $[\text{Bmim}]\text{PF}_6$ can be separated from other components via distillation, and $[\text{Bmim}]\text{PF}_6$ can be reused. In industry, CH_3OH , CH_3CN and H_2O can be separated by a separation equipment (Ref, Patent CN103386211), which consists mainly of a separating and heating unit, and distillation unit, a fine stripping section, cooling towers, a rectification column, condenser, and a fluid transfer tube. The separation efficiency of methanol can reach to ~98% for practical application. Here, we would like to emphasize that, while this paper focuses on reporting the interesting results for the catalytic reaction, the

separation of the product is crucial for practical application, as mentioned by the referee, which should be studied and optimized systemically in case the electrochemical method to reduce CO₂ is used in industry in future.

2. The comparisons in Fig. 3a and S14 do not take into account surface area. It is very difficult to assess differences in activity across materials when the electrode morphologies/surface areas are significantly different.

Response: We thank the referee for the comment. As suggested by the referee, to better evaluate the intrinsic activity of various electrodes, we compared the current densities of various electrodes *vs* electrochemical active surface areas (ECSA) determined using the reported method⁵⁰. The results are shown in Supplementary Fig. 24 and Supplementary Table 5 and the ECSA normalized current densities also are given in Supplementary Table 5. The results indicate clearly that the Cu_{1.63}Se(1/3) is the most efficient catalyst. Some statements on this have been added in revised manuscript as following.

“The catalytic performance was also evaluated using the electrochemical active surface areas (ECSA) determined by reported method⁵⁰. Results in Supplementary Fig. 24 and Supplementary Table 5 show that the formation rate of methanol over Cu_{1.63}Se(1/3) was intrinsically higher than that on the other catalysts.” Please see Page 9 in the revised manuscript.

“The ECSA values of all electrodes were evaluated by cyclic voltammetry (CV) using the ferri-/ferrocyanide redox couple ([Fe(CN)₆]^{3-/4-}) as a probe⁵⁰. The CV curves were obtained in a N₂-saturated 5 mM K₄Fe(CN)₆/0.1 M KCl solution including a counter anode (platinum gauzes), and a reference electrode (Ag/AgCl with saturated KCl). According to the Randles-Sevcik equation⁵⁰, the values of ECSA were obtained.” Please see Page 15 in the revised manuscript.

3. The LSVs in Fig. 3b are used to probe “surface adsorption” of sulfate. It is not clear how to interpret the data. What does the current correspond to? These are not

adsorption features and represent either oxidation of the electrode itself or water. In any case, the adsorption of sulfate under strong anodic potential has nothing to do with the adsorption of CO₂ reduction intermediates.

Response: We thank the referee for the comment. We agree with the referee that the property of sulfate is different from that of reduced CO₂ radical anion and the adsorption of sulfate is not equal to the adsorption of reduced CO₂ radical anion quantitatively. However, sulfate is often used as the analogs of the CO₂⁻ ion to qualitatively compare the variation of binding energy for CO₂⁻ intermediates on various materials^{21, 49, 51} because it is very difficult to find better analogs considering the various factors, such as chemical stability at the electrochemical reaction condition. Therefore, we used sulfate in this work, and the results support our conclusion. Besides sulfate, we also tried to use both formate and acetate, but they are not stable at the experimental condition. However, this is only one of experimental evidences to explain why Cu_{1.63}Se(1/3) showed excellent performance. If the referee insists on removing this, we would like to do this because the conclusion of the paper is the same without these results. In revised manuscript, we have discussed this as following.

“It is known that smaller overpotential of hydroxyls and sulfate adsorption indicates larger binding energy on intermediates⁵¹. Therefore, hydroxyls and sulfates (or bisulfates) can be used as the analogs of the CO₂⁻ ion to study the binding energy of intermediates. In this work, we studied the binding affinity for CO₂⁻ intermediates on various electrodes using sulfate, and the method is similar to that in the previous reports^{21, 49, 51}. Fig. 3b shows LSVs of the sulfate adsorption peaks on different catalysts in 0.1 M sulfuric acid. It can be known that the sulfate binds more strongly on Cu_{1.63}Se(1/3).” Please see page 9 in the revised manuscript.

4. The authors propose hydrogen bonding between CO₂ and the Bmim⁺ electrolyte. CO₂ is a terrible hydrogen bond acceptor and bmim is a rather weak donor. What is the evidence for this interaction?

Response: We thank the referee for the comment. We agree with the referee that the

hydrogen bonding between CO₂ and the Bmim⁺ electrolyte was very weak. It was reported that CO₂ can form complex with the imidazolium cations to lower the overpotentials², since there exists weak Lewis acid-base interaction between the CO₂ and the anions of the ionic liquids (Ref, Kazarian, S. G., Brian J. B. & Thomas. *Chem. Commun.* **20**, 2047-2048, (2000).). In the revised manuscript, we removed “hydrogen bonding between CO₂ and the Bmim⁺”, and discussed the interaction of CO₂ and IL [Bmim]PF₆ by “The electrolyte containing ionic liquids and CO₂ formed complex CO₂-[Bmim]PF₆, which can enhance the CO₂ concentration in electrolyte and transport CO₂ to the catalyst surface to improve further reduction of CO₂².” Please see page 11 in the revised manuscript.

Reviewers' comments:

Reviewer #1 (Remarks to the Author):

Yang et al have provided a very interesting report on the use of CuSe nanoparticles as electrode materials for the selective electrochemical conversion of CO₂ to methanol in ternary ionic liquid mixtures. Cu is a very good electrode material for CO₂ reduction, though without much selectivity. This report therefore is very interesting in terms of the apparent ability of non-stoichiometric CuSe nanoparticles to orchestrate a complex series of 6e⁻, 6H⁺ steps. The materials are well-characterized, and product identify supported by a number of techniques, including IR (added in revised version).

Having considered the revised manuscript in light of the authors' additions and reviewers comments, I still believe this is novel, exciting work, and warrants communication to Nat. Comm.'s broad readership as this fundamental study suggests opportunities now for engineering to take over to potentially lead to practical systems.

In terms of the revisions, the authors' have largely addressed my concerns from their initial submission. I would recommend acceptance pending addressing the following minor revisions. Also, perhaps as a result of the revision, the manuscript feels repetitive in parts and might benefit from careful editing.

Title: change "toward" to "to"

Line 48: "were" not "was"

Line 62: "synthesized"

Line 89: "nanocatalysts"

Line 122: "nuclear magnetic resonance spectroscopy"

Line 170: This sentence should include a reference. Reference 43?

Line 212-213 and Fig4a: the adsorption of CO₂ to the surface followed by reduction is shown as a concerted process in Fig4a, but described with some certainty as a step-wise process in lines 212-213. Please clarify.

Line 225: "rate-limiting". Figure 4b does not appear to show barriers. Were these (transition states) calculated? Is something known about the relative barriers from DFT? Please discuss if so.

Reviewer #3 Comments and Evaluation of Response

Demonstrating the significance of an electrochemical process whose merits rest on reporting high current density and Faradaic efficiency (less so overpotential, really) requires careful attention to the practical implementation of a system.

The work does appear technically sound, and the answer provided in the response to the reviewer comments addresses issues with respect to practical implementation (specifically, the importance of maximizing FE and current density, and the impossibility of really addressing overall cell potential).

To properly address the fact that a reader may leave with a first impression that this is not a practical system for fuel formation, these arguments given in response to the reviewer comment should be evident to some extent in the text. That is, can you explain why, given the opportunity to address cell potential, you focus on maximizing FE and current density by using CuSe (novel approach) in mixed organic/aqueous media (not a novel approach) despite the organic solvents

likely large contribution to iR drop? Can you explicitly describe for readers who might suspect low process efficiency a rationale for why given the likelihood of high iR it's still worth it to use your system? Same comment with respect to separation of products.

Comment #2 appears to be well-addressed (surface area).

With respect to comment #3, while a justification for the use of sulfate is given, the authors have not responded to the question of what the current responds to (why adsorption and not oxidation?) Without addressing this properly, the example should likely be removed.

Comment #4 is addressed satisfactorily.

Reviewer #2 (Remarks to the Author):

Through carefully reading all the comments of three reviewers, the replies to all reviewers, and the revised manuscript and supplementary information, I noted that the authors have made major revisions and conducted additional experimental measurements and DFT calculations to meet the suggestions and criticisms raised by the three reviewers. After adding more structural characterizations, such as HRETM, XPS, XAFS, TG measurements, the structure of the as-synthesized $\text{Cu}_{1.63}\text{Se}_{(1/3)}$ electrocatalyst is more clear. To investigate the reaction mechanism for electrocatalytic CO_2 reduction, FTIR measurements and other control experiments in the presence of formic acid, CO, and formaldehyde were conducted. Necessary DFT calculations were also carried out, yielding the free energy diagram for various intermediates.

The authors have addressed most of the questions raised by Reviewer #3. The first concern of Reviewer #3 is about the significance of the result. In my opinion, this work reports the electrochemical reduction of CO_2 to methanol using copper selenide as the catalyst is of broad interest to researchers working on fundamental issues in materials, energy, and catalysis, although the industry application of this reaction is still a little far. In order to respond to Reviewer #3's concern on the electrode surface areas, the authors have also evaluated the intrinsic activity of various electrodes through comparing the current densities of various electrodes vs. electrochemical active surface areas (ECSA).

But there are still some unanswered or not-well-addressed questions. The first is on the efficiency of the cell with an overall voltage of ~ 3.5 eV (Question 1 by Reviewer #3). The authors mentioned many times that the $\text{Cu}_{1.63}\text{Se}_{(1/3)}$ is the most efficient catalyst, at least a brief comment on the cell efficiency should be made. Reviewer #3's another concern is on why to use the adsorption of sulfate to analogize the adsorption of CO_2 radical anion (Question 3 of Reviewer #3). I agree with Reviewer #3 that surface adsorption of sulfate does not directly correlate with the adsorption features of the electrode itself or water. Hence, it is better to remove this result from this work, and try their best to give a reasonable experimental explanation.

In addition, a careful inspection of the EXAFS results reveals that the fitting results are quite questionable. The FT curves in Fig. S25(c) indicate the same Cu-Se peak positions for the three sample, in contrast to the large difference of the extracted Cu-Se bond lengths in Table 6. Besides, the extracted coordination numbers for the three samples are all around 4.0, although their FT peak intensity differs significantly. This coordination number is inconsistent with the structure models used for DFT calculations. As the free energies of the reaction intermediates rely on the adsorption configuration, this inconsistency may affect the correctness of the mechanistic interpretation.

To summarize, if these above mentioned issues could be satisfactorily clarified, I consider this work is worthy of publication in Nature Communications.

Responses to the comments and revisions made

Referee 1

Remarks to the Author

Yang et al have provided a very interesting report on the use of CuSe nanoparticles as electrode materials for the selective electrochemical conversion of CO₂ to methanol in ternary ionic liquid mixtures. Cu is a very good electrode material for CO₂ reduction, though without much selectivity. This report therefore is very interesting in terms of the apparent ability of non-stoichiometric CuSe nanoparticles to orchestrate a complex series of 6e⁻, 6H⁺ steps. The materials are well-characterized, and product identify supported by a number of techniques, including IR (added in revised version).

Having considered the revised manuscript in light of the authors' additions and reviewers' comments, I still believe this is novel, exciting work, and warrants communication to Nat. Comm.'s broad readership as this fundamental study suggests opportunities now for engineering to take over to potentially lead to practical systems.

In terms of the revisions, the authors have largely addressed my concerns from their initial submission. I would recommend acceptance pending addressing the following minor revisions. Also, perhaps as a result of the revision, the manuscript feels repetitive in parts and might benefit from careful editing.

Response: We thank the referee for the comment. On the basis of the comment, we have reedited the contents and deleted the repetitive statements. We also corrected the minor mistakes after careful editing in the revised manuscript.

1. Title: change “toward” to “to”.

Response: We thank the referee for the comment. The “toward” has been changed into “to” in the revised manuscript. (Please see the title of the revised manuscript).

2. Line 48: “were” not “was”.

Response: We thank the referee for the comment. In the revised manuscript, we have

changed “was” into “were” in the Line 48. (Please see Page 3 in the revised manuscript).

3. Line 62: “synthesized”.

Response: We thank the referee very much for the comment. According the suggestion of referee, “synthesize” has been modified to “synthesized”. (Please see Page 3 in the revised manuscript).

4. Line 89: “nanocatalysts”.

Response: We thank the referee for the comment. The “nanocayalysts” which had a spelling mistake has been changed to “nanocatalysts” in the revised manuscript. (Please see Page 5 in the revised manuscript).

5. Line 122: “nuclear magnetic resonance spectroscopy”.

Response: We thank the referee for the comment. In the revised manuscript, we have changed “nuclear magnetic resonance” into “nuclear magnetic resonance spectroscopy”. (Please see Page 7 in the revised manuscript).

6. Line 170: This sentence should include a reference. Reference 43?

Response: We thank the referee for the comment. To better illustrate this sentence, we have added the reference as following.

“which resulted partially from the difference of the interaction between CO₂ and the anions of the ILs⁵⁵.” (Please see Page 9 in the revised manuscript).

7. Line 212-213 and Fig 4a: the adsorption of CO₂ to the surface followed by reduction is shown as a concerted process in Fig 4a, but described with some certainty as a step-wise process in lines 212-213. Please clarify.

Response: We thank the referee for the comment. According to the advice of referee, the adsorption of CO₂ to the surface followed by reduction is a concerted process. To better explain the phenomenon, we have modified the statement as following.

“In the initial stage of the reduction, the electrolyte containing ionic liquids and CO₂ formed complex CO₂-[Bmim]PF₆, which can enhance the concentration of CO₂ in electrolyte and transport of CO₂ to the catalyst surface to improve further transform of CO₂ into CO₂^{•-}.⁵⁵” (Please see Page 11 in the revised manuscript).

8. Line 225: “rate-limiting”. Figure 4b does not appear to show barriers. Were these (transition states) calculated? Is something known about the relative barriers from DFT? Please discuss if so.

Response: We thank the referee for the comment. To compare the differences among Cu_{1.63}Se(1/3), Cu₂Se and CuSe, DFT calculations were conducted and the results are shown in Figure 4b and Supplementary Figs. 31 and 32. The step of *CO reduction to *CHO was an endothermic and rate-limiting step according to the largest difference of free energy (0.56 eV for Cu_{1.63}Se(1/3), 1.14 eV for Cu₂Se and 1.25 eV for CuSe). In addition, based on the Brønsted-Evans-Polanyi (BEP) relationship (Ref, *Trans. Faraday Soc.* **1938**, 34, 11; *J. Phys. Chem. C*, **2008**, 112, 1308), the reaction barrier has a linear relationship to the reaction energy, and this method does not affect the results of DFT calculation considerably. In the revised manuscript, we have modified the related sentences as following.

“Based on the Brønsted-Evans-Polanyi (BEP) relationship^{60, 61}, the reaction barrier has a linear relationship to the reaction energy, and it can also be seen that the step of *CO reduction to *CHO was an endothermic and rate-limiting step since the highest energy potential (0.56 eV) is needed in this step.” (Please see Page 12 in the revised manuscript).

Referee 2

Remarks to the Author

Through carefully reading all the comments of three reviewers, the replies to all reviewers, and the revised manuscript and supplementary information, I noted that the authors have made major revisions and conducted additional experimental measurements and DFT calculations to meet the suggestions and criticisms raised by

the three reviewers. After adding more structural characterizations, such as HRETm, XPS, XAFS, TG measurements, the structure of the as-synthesized $\text{Cu}_{1.63}\text{Se}(1/3)$ electrocatalyst is more clear. To investigate the reaction mechanism for electrocatalytic CO_2 reduction, FTIR measurements and other control experiments in the presence of formic acid, CO, and formaldehyde were conducted. Necessary DFT calculations were also carried out, yielding the free energy diagram for various intermediates.

The authors have addressed most of the questions raised by Reviewer #3. The first concern of Reviewer #3 is about the significance of the result. In my opinion, this work reports the electrochemical reduction of CO_2 to methanol using copper selenide as the catalyst is of broad interest to researchers working on fundamental issues in materials, energy, and catalysis, although the industry application of this reaction is still a little far. In order to respond to Reviewer #3's concern on the electrode surface areas, the authors have also evaluated the intrinsic activity of various electrodes through comparing the current densities of various electrodes vs. electrochemical active surface areas (ECSA).

Response: We thank the referee for the comment. On the basis of the comment, we have made point-to-point modifications as following.

1. But there are still some unanswered or not-well-addressed questions. The first is on the efficiency of the cell with an overall voltage of ~ 3.5 eV (Question 1 by Reviewer #3). The authors mentioned many times that the $\text{Cu}_{1.63}\text{Se}(1/3)$ is the most efficient catalyst, at least a brief comment on the cell efficiency should be made.

Response: We thank the referee for the comment. According to the advice of referee, we have added some statements about cell voltage and cell efficiency, and a brief comment has been made in the revised manuscript.

First, we calculated the cell voltage and cell efficiency using the corresponding reported method (Ref, *Nat. Catal.*, **2018**, 1, 11; *Proc. Natl. Acad. Sci. U. S. A.* **2016**, 113, 5526; *Angew. Chem. Int. Ed.*, **2018**, 57, 6883.). Second, we have given a systematic summary about cell voltage of CO_2 reduction to different products

reported in the literature (Supplementary Table 3). Third, a brief comment on the above results has been made. The overall cell voltage of our system at optimized reaction condition calculated using the reported method (Ref, *Nat. Catal.*, **2018**, 1, 11; *Proc. Natl. Acad. Sci. U. S. A.* **2016**, 113, 5526.) is 2.67 V, which is in the range of cell voltage values (2.2-3.7 V, Supplementary Table 3). The highest energy efficiency was 61.7% was obtained at cell voltage of 2.67 V. In the revised manuscript, as suggested by the referee, we have discussed this as following.

“The cell voltage is an important factor for practical application, which depends mainly on the performances of the electrocatalysts. In this study, we calculated the cell voltage using the reported method^{46, 47}, and the cell voltage of our system was 2.67 V, which is in the range of reported values (2.2-3.7 V, Supplementary Table 3). In Supplementary Fig. 12, the FE for methanol production increased with the cell voltage to reach the maximum value of 77.6% at 2.67 V. We also calculated the energy efficiency (EE) for methanol production at different cell voltages using the reported method⁴⁸, and the results are given in Supplementary Fig. 12. The EE exhibited a similar tendency to the FE of methanol with variation of the cell voltage. Furthermore, the highest EE was 61.7% at the optimized cell voltage of 2.67 V.” Please see Page 7 in the revised manuscript.

The above results are given in Supplementary Fig. 12, and the methods to calculate cell voltage and cell efficiency are also shown Page 2 in the revised Supplementary Information, together with the corresponding references.

2. Reviewer #3's another concern is on why to use the adsorption of sulfate to analogize the adsorption of CO₂ radical anion (Question 3 of Reviewer #3). I agree with Reviewer #3 that surface adsorption of sulfate does not directly correlate with the adsorption features of the electrode itself or water. Hence, it is better to remove this result from this work, and try their best to give a reasonable experimental explanation.

Response: We thank the referee again for the comment. According to the advice of referee, the contents about surface adsorption of sulfate have been removed from the

manuscript. The control experiments in the manuscript can sufficient illustrate the role of Se in the catalysts. That is, when using Cu, CuO, Cu₂O, CuS, Cu₂S, CuSe and Cu₂Se as catalysts, both current density and FE for methanol over Se-free catalysts were obvious lower (Fig. 3 and Supplementary Fig. 24). The results suggested that the Cu and Se in the catalysts cooperated very well for the formation of methanol. In other words, the capacity of electroreduction CO₂ to methanol was enhanced when O or S atom was replaced by Se atom in the catalysts. In addition, when CuSe or Cu₂Se was utilized as the catalysts, both current density and FE were much lower than that over Cu_{1.63}Se(1/3). The results in Supplementary Fig. 25 and Supplementary Table 6 also show that the formation rate of methanol over Cu_{1.63}Se(1/3) was intrinsically higher than that on the other catalysts. Thus, on the basis of above results, we can deduce that Se in the catalysts is crucial for efficient CO₂ reduction to methanol. In summary, the results can be explained without the results of sulfate adsorption. The above explanation was given in the previous version of the manuscript and is not changed in this version (Please see Page 10).

3. In addition, a careful inspection of the EXAFS results reveals that the fitting results are quite questionable. The FT curves in Fig. S26(c) indicate the same Cu-Se peak positions for the three sample, in contrast to the large difference of the extracted Cu-Se bond lengths in Table 7. Besides, the extracted coordination numbers for the three samples are all around 4.0, although their FT peak intensity differs significantly. This coordination number is inconsistent with the structure models used for DFT calculations. As the free energies of the reaction intermediates rely on the adsorption configuration, this inconsistency may affect the correctness of the mechanistic interpretation.

Response: We are very grateful to the referee for pointing out the mistake. In the last time, we considered that coordination number (CN) was 4 for both CuSe and Cu₂Se in the fitting process. After carefully checking, we find that the CN in the CuSe and Cu₂Se should be 6 and 4, respectively. Thus, we re-fitted data, and the results are shown in Supplementary Figs. 27-30 and Supplementary Table 7. The new data

agreed with the crystal structure for the DFT calculations. The statements were shown in the revised manuscript as following.

“It can be seen that the coordination number in $\text{Cu}_{1.63}\text{Se}(1/3)$ was smaller than that in CuSe and Cu_2Se (Supplementary Figs. 27-30 and Supplementary Table 7). Thus, there existed unsaturated Se atom in the $\text{Cu}_{1.63}\text{Se}(1/3)$, which may enhance the performance for CO_2 electroreduction.” (Please see Page 11 in the revised manuscript).

To summarize, if these above mentioned issues could be satisfactorily clarified, I consider this work is worthy of publication in Nature Communications.

Response: We thank very much the referee for the comment.

Referee 3

Remarks to the Author

1. Demonstrating the significance of an electrochemical process whose merits rest on reporting high current density and Faradaic efficiency (less so overpotential, really) requires careful attention to the practical implementation of a system.

The work does appear technically sound, and the answer provided in the response to the reviewer comments addresses issues with respect to practical implementation (specifically, the importance of maximizing FE and current density, and the impossibility of really addressing overall cell potential).

To properly address the fact that a reader may leave with a first impression that this is not a practical system for fuel formation, these arguments given in response to the reviewer comment should be evident to some extent in the text. That is, can you explain why, given the opportunity to address cell potential, you focus on maximizing FE and current density by using CuSe (novel approach) in mixed organic/aqueous media (not a novel approach) despite the organic solvents likely large contribution to iR drop? Can you explicitly describe for readers who might suspect low process efficiency a rationale for why given the likelihood of high iR it's still worth it to use your system? Same comment with respect to separation of products.

Response: We thank the reviewer for the very instructive comment. We have addressed all the concerns, as can be known from the answers to the comments as discussed in the following.

1) About practical application

According to the comment of the referee, in the Conclusion (Summary) section, we have added “Despite the catalytic system is far from industrial production, it is still very interesting that $\text{Cu}_{1.63}\text{Se}(1/3)$ nanocatalysts can yield highest current density up to date at very high Faradaic efficiency.” Please see Page 12 in the revised manuscript.

2) About the cell potential.

As mentioned by the referee, cell voltage is an important factor for practical application. In the revised manuscript, we calculated cell voltage of our system using the reported method (Ref, *Nat. Catal.*, **2018**, 1, 11; *Proc. Natl. Acad. Sci. U. S. A.* **2016**, 113, 5526.). The cell voltage of our system at optimized reaction condition is 2.67 V. We also give a table (Supplementary Table 3) to list the cell voltage values. We found in the literature for CO_2 reduction to different products (most papers on CO_2 reduction do not report cell voltage, and they pay attention to current density, Faradaic efficiency, and overpotential), and the values of cell voltage are in the range of 2.2-3.7 V (Supplementary Table 3). In the revised manuscript, as suggested by the referee, we have discussed this as following.

“The cell voltage is an important factor for practical application, which depends mainly on the performances of the electrocatalysts. In this study, we calculated the cell voltage using the reported method^{46, 47}, and the cell voltage of our system was 2.67 V, which is in the range of reported values (2.2-3.7 V, Supplementary Table 3).” Please see Page 7 in the revised manuscript.

The above results are given in Supplementary Fig. 12, and the method to calculate cell voltage is also shown Page 2 in the revised Supplementary Information, together with the corresponding references.

3) Can you explain why, given the opportunity to address cell potential, you focus on maximizing FE and current density by using CuSe (novel approach) in mixed organic/aqueous media (not a novel approach) despite the organic solvents likely large contribution to iR drop? Can you explicitly describe for readers who might suspect low process efficiency a rationale for why given the likelihood of high iR it's still worth it to use your system?

Answer: We agree with the referee that the organic solvents generate relatively large iR drop. Many researchers have reported the electroreduction CO₂ in aqueous electrolyte as catholyte. Although water is cheaper, easily available, and greener, the evolution of hydrogen in aqueous electrolyte is much easier than reduction of CO₂, and thus the Faradaic efficiency is usually low, especially at larger current density. The combination of organic solvents and ionic liquids as electrolyte has some obvious advantages. Firstly, this kind of electrolyte can accelerate the CO₂ reduction rate (i.e. current density and Faradaic efficiency) by increasing the adsorption rate of CO₂ comparing with aqueous electrolyte (Ref, *Ind. Eng. Chem. Res.* **2006**, *45*, 8180; *ACS Catal.* **2015**, *5*, 6440). Secondly, compared with aqueous electrolyte, the use of this kind of electrolyte can provide more opportunity to produce various products (Ref, *J. Am. Chem. Soc.* 2014, **136**, 8361; *Science*, **2011**, *334*, 643; *Angew. Chem. Int. Ed.* **2016**, *55*, 6771; *Angew. Chem. Int. Ed.* **2016**, *55*, 9012; *Angew. Chem. Int. Ed.* **2018**, *57*, 2427). Therefore, despite organic system may contribute to iR drop, we believe that this system is suitable because multi-electron/proton coupling steps can occur in this system to obtain valuable products with high current density and high selectivity. In our work, combination of copper selenide and ionic liquid-CH₃CN-H₂O can yield highest current density up to date for methanol generation at very high Faradaic efficiency (Supplementary Table 1). Meanwhile, based on the above result, the energy efficiency (EE) for methanol production at different cell voltages was also calculated using the reported method (Ref, *Angew. Chem. Int. Ed.*, **2018**, *57*, 6883.). We found the EE exhibited a similar tendency to the FE of methanol with variation of the cell voltage. Furthermore, the highest EE was 61.7% at the optimized cell voltage of 2.67

V. Despite the catalytic system is far from industrial production, it is still very interesting that $\text{Cu}_{1.63}\text{Se}(1/3)$ nanocatalysts can yield highest current density up to date at very high Faradaic efficiency.

In this revised manuscript, we have discussed this as following.

“We also calculated the energy efficiency (EE) for methanol production at different cell voltages and the results are given in Supplementary Fig. 12. The EE exhibited a similar tendency to the FE of methanol with variation of the cell voltage. Furthermore, the highest EE was 61.7% at the optimized cell voltage of 2.67 V.” Please see Page 7 in the revised manuscript.

“Compared with aqueous electrolyte, combination of organic solvents and ionic liquids as electrolytes has obvious advantages. For example, they can accelerate catalytic performance of CO_2 reduction by increasing the adsorption rate of CO_2 ^{51, 52}, and they provide more opportunity to produce various valuable products^{1, 2, 8, 53, 54}.” Please see Page 9 in the revised manuscript.

4) About separation of products.

We also agree with the referee that separation of a polar product is a very important factor for practical application. As discussed above, we mainly report our discovery that $\text{Cu}_{1.63}\text{Se}(1/3)$ nanocatalyst is an outstanding electrocatalyst, and the separation of methanol is not studied in this work. Here we would like to discuss this briefly in response to the comment. We used $[\text{Bmim}]\text{PF}_6/\text{CH}_3\text{CN}/\text{H}_2\text{O}$ ternary electrolyte. $[\text{Bmim}]\text{PF}_6$ is not volatile, while CH_3OH , CH_3CN and H_2O are volatile. Thus, the $[\text{Bmim}]\text{PF}_6$ can be separated from other components via distillation. $[\text{Bmim}]\text{PF}_6$ can be reused. In industry, CH_3OH , CH_3CN and H_2O can be separated by a separation equipment (Ref, Patent CN103386211), which consists mainly of a separating and heating unit, and distillation unit, a fine stripping section, cooling towers, a rectification column, condenser, and a fluid transfer tube. The separation efficiency of methanol can reach to ~98% for practical application. Here, we would like to emphasize that, while this paper focuses on reporting the interesting results for

the catalytic reaction, the separation of the product is crucial for practical application as mentioned by the referee, which should be studied and optimized systemically in case the electrochemical method to reduce CO₂ is used in industry in the future. In the revised manuscript, we have discussed this as following “The separation of the reaction mixture is crucial for practical application. Although this is out of the scope of this work, we would like to discuss this very briefly. For this system, the boiling point of [Bmim]PF₆ is much higher than that of CH₃OH, CH₃CN and H₂O, and the IL in the system can be separated via distillation. Meanwhile, the method to separate ternary mixture consisting of CH₃OH, CH₃CN and H₂O has been reported⁵⁶.” Please see Page 9 in the revised manuscript.

2. Comment #2 appears to be well-addressed (surface area).

Response: We thank the referee for the comment.

3. With respect to comment #3, while a justification for the use of sulfate is given, the authors have not responded to the question of what the current responds to (why adsorption and not oxidation?) Without addressing this properly, the example should likely be removed.

Response: We thank the referee for the comment. According to the advice of referee, we have removed the statements about the sulfate adsorption in the revised manuscript.

Comment #4 is addressed satisfactorily.

Response: We thank the referee for the comment.

REVIEWERS' COMMENTS:

Reviewer #1 (Remarks to the Author):

The authors have addressed all my concerns satisfactorily, as well, it would seem, as those of the other reviewers. I have included minor comments below, but feel the manuscript is certainly exciting and suitable in its present form for publication.

Abstract - "copper selenide nanocatalysts HAVE"

Page 2, line 38 - "could be used as electrocatalyst TO promote the reaction"

Page 2, line 40 - "reported up to date could reach relatively high current density ($> 10 \text{ mA cm}^{-2}$) and selectivity ($> 50\%$)," Why these particular benchmarks?

Page 3, line 53 - "In addition, they are low-cost COMPARED with" (remove ALSO)

Page 5, line 82 - "TG curve is shown in Supplementary Fig. 3." TG is not defined. Also change "if existed" to "if incorporated into the nanoparticles"

Page 10, line 190 - "the capacity of electroreduction OF CO₂ to"

Page 11, line 211 - Likely should remove the statement "CO₂ formed complex CO₂-[Bmim]PF₆," as the reference does not indicate formation of any IL-CO₂ complex (as discussed in previous rounds of reviews). Also, Figure 4 shows that CO₂- if formed is formed as adsorbed CO₂-. This should be made clear in the discussion in lines 211-216.

Page 12, line 225 - I would feel more comfortable if the statement about rate-limiting step was qualified as barriers are not discussed. Perhaps "was an endothermic and LIKELY rate-limiting step"

Reviewer #2 (Remarks to the Author):

In the revised manuscript NCOMMS-18-18231B, the authors have calculated the cell voltage and the cell efficiency in Supplementary Figure 12, and added some useful discussions on the cell efficiency according to my suggestions. Moreover, the authors have used the control experiments to illustrate the role of Se in the catalysts. The authors have also re-fitted the EXAFS data in Supplementary Figures 27-30, and now the fitting results in Supplementary Table 7 are reasonable. In summary, the authors have supplemented necessary experimental results to support their conclusions and clarified most of the questions and queries raised in the previous round of review, and now I recommend the publication of this work in Nature Communications.

Responses to the comments and revisions made

Referee 1

Remarks to the Author

The authors have addressed all my concerns satisfactorily, as well, it would seem, as those of the other reviewers. I have included minor comments below, but feel the manuscript is certainly exciting and suitable in its present form for publication.

Response: We thank the referee very much for the comment. We have addressed all the points raised by the reviewer, as summarized below.

1. Abstract - "copper selenide nanocatalysts HAVE"

Response: We thank the referee for the comment. The sentence "copper selenide nanocatalysts has" has been changed into "copper selenide nanocatalysts have" in the revised manuscript. (Please see the abstract of the revised manuscript).

2. Page 2, line 38 - "could be used as electrocatalyst TO promote the reaction"

Response: We thank the referee very much for the comment. The sentence "could be used as electrocatalyst promote the reaction" has been modified to "could be used as electrocatalyst to promote the reaction". (Please see Page 2 in the revised manuscript).

3. Page 2, line 40 - "reported up to date could reach relatively high current density ($> 10 \text{ mA cm}^{-2}$) and selectivity ($> 50\%$)," Why these particular benchmarks?

Response: We thank the referee for the comment. As suggested by the reviewer, we have removed the particular benchmarks in the revised manuscript.

"and only a few catalysts reported up to date could reach relatively high current density and selectivity" (Please see Page 2 in the revised manuscript).

4. Page 3, line 53 - "In addition, they are low-cost COMPARED with" (remove ALSO)

Response: We thank the referee for the comment. In the revised manuscript, the sentence "In addition, they are also low-cost comparing with" has been modified to

“In addition, they are low-cost compared with” in the revised manuscript. (Please see Page 3 in the revised manuscript).

5. Page 5, line 82 - "TG curve is shown in Supplementary Fig. 3." TG is not defined. Also change "if existed" to "if incorporated into the nanoparticles"

Response: We thank the referee for the comment. According to the advice of referee, we have added definition of TG. The statements in the revised manuscript have been modified as following.

“Thermogravimetry (TG) curve is shown in Supplementary Fig. 3. The weight loss of DETA (if incorporated into the nanoparticles) should occur at 280 °C-580 °C³⁶.” (Please see Page 4 in the revised manuscript).

6. Page 10, line 190 - "the capacity of electroreduction OF CO₂ to"

Response: We thank the referee for the comment. The sentence “the capacity of electroreduction CO₂ to” has been changed into “the capacity of electroreduction of CO₂ to” in the revised manuscript. (Please see Page 9 in the revised manuscript).

7. Page 11, line 211 - Likely should remove the statement "CO₂ formed complex CO₂-[Bmim]PF₆," as the reference does not indicate formation of any IL-CO₂ complex (as discussed in previous rounds of reviews). Also, Figure 4 shows that CO₂⁻ if formed is formed as adsorbed CO₂⁻. This should be made clear in the discussion in lines 211-216.

Response: We thank the referee for the comment. According to the referee’s suggestion, we have modified the statements in the revised manuscript as following.

“the electrolyte containing ionic liquids can enhance the concentration of CO₂ in electrolyte and transport of CO₂ to the catalyst surface to improve further transform of CO₂ into adsorbed CO₂⁻ ^{2, 55}. The adsorbed CO₂⁻ could bind with...” (Please see Page 9 in the revised manuscript).

8. Page 12, line 225 - I would feel more comfortable if the statement about

rate-limiting step was qualified as barriers are not discussed. Perhaps "was an endothermic and LIKELY rate-limiting step"

Response: We thank the referee for the comment. The statement "was an endothermic and rate-limiting step" has been changed into "was an endothermic and likely rate-limiting step" in the revised manuscript. (Please see Page 10 in the revised manuscript).

Referee 2

Remarks to the Author

In the revised manuscript NCOMMS-18-18231B, the authors have calculated the cell voltage and the cell efficiency in Supplementary Figure 12, and added some useful discussions on the cell efficiency according to my suggestions. Moreover, the authors have used the control experiments to illustrate the role of Se in the catalysts. The authors have also re-fitted the EXAFS data in Supplementary Figures 27-30, and now the fitting results in Supplementary Table 7 are reasonable. In summary, the authors have supplemented necessary experimental results to support their conclusions and clarified most of the questions and queries raised in the previous round of review, and now I recommend the publication of this work in Nature Communications.

Response: We thank the referee very much for the comment.